# The environmental sustainability of digital content consumption

Robert Istrate [1,2], Victor Tulus [1], Robert N. Grass [1], Laurent Vanbever[3], Wendelin J. Stark[1] & Gonzalo Guillén-Gosálbez [1] ✉

Internet access has reached 60% of the global population, with the average user spending over 40% of their waking life on the Internet, yet the environmental implications remain poorly understood. Here, we assess the environmental impacts of digital content consumption in relation to the Earth's carrying capacity, finding that currently the global average consumption of web surfing, social media, video and music streaming, and video conferencing could account for approximately 40% of the per capita carbon budget consistent with limiting global warming to 1.5 °C, as well as around 55% of the per capita carrying capacity for mineral and metal resources use and over 10% for five other impact categories. Decarbonising electricity would substantially mitigate the climate impacts linked to Internet consumption, while the use of mineral and metal resources would remain of concern. A synergistic combination of rapid decarbonisation and additional measures aimed at reducing the use of fresh raw materials in electronic devices (e.g., lifetime extension) is paramount to prevent the growing Internet demand from exacerbating the pressure on the finite Earth's carrying capacity.

Internet access has already reached more than 5 billion people (about 60% of the global population)[1], leading to a global data traffic of 3.4 ZB in 2021, a remarkable 440% growth since 2015[2]. At present, the average Internet user spends approximately seven hours per day online (>40% of their waking life), with social media and video streaming being the most widely consumed digital services (over two hours per day)[1]. Despite the ubiquity of the Internet in our lives, its environmental footprint started to gain public interest only recently, partially fuelled by studies highlighting the substantial energy consumption of the information and communication technology (ICT) sector[3–5].

Prior studies have analysed the energy consumption of global data centres[3,6–8] and data transmission networks[9–12], finding that they collectively account for 2–3% of the global electricity consumption[2,6]. Furthermore, a number of studies have specifically assessed the environmental impacts of the ICT sector[5,13–15] and/or its essential components (data centres[16–18], data transmission[19], and end-user devices[20,21]), as well as the provided digital services (e.g., social media[22], video streaming[23–26], virtual conferences[27–31], artificial inteligence[32], cryptocurrencies[33–35], and online advertising[36]). For example, it was shown that the global ICT sector (i.e., data centres, data transmission networks, and end-user devices) emitted 1.0–1.7 Gt $CO_2$-eq in 2020, considering both operational and embodied emissions[37]. This is equivalent to 1.8–2.8% of global anthropogenic greenhouse gas (GHG) emissions, surpassing the GHG emissions of Australia or France[38,39].

Notwithstanding these studies of the environmental impacts associated with the ICT sector, a bottom-up analysis of Internet consumption considering the users' consumption patterns of different digital contents is still lacking –in contrast to basic human needs such as food and transport, which have been shown to contribute ~20% each to the carbon footprint of an average individual[40,41]–. Moreover, it is important to address the fundamental question of whether (and to

[1]Institute for Chemical and Bioengineering, Department of Chemistry and Applied Biosciences, ETH Zürich, Vladimir-Prelog-Weg 1, 8093 Zürich, Switzerland. [2]Institute of Environmental Sciences (CML), Leiden University, Einsteinweg 2, 2333 CC Leiden, The Netherlands. [3]Computer Engineering and Networks Laboratory, Department of Information Technology and Electrical Engineering, ETH Zürich, Gloriastrasse 35, 8092 Zürich, Switzerland. ✉e-mail: gonzalo.guillen.gosalbez@chem.ethz.ch

what extent) the growing Internet consumption could pose a challenge to our ability to shift to environmentally sustainable lifestyles consistent with the Earth's ecological capacity[42–44].

Here we quantify the environmental impacts of digital content consumption encompassing all the necessary infrastructure linked to the consumption patterns of an average user. By applying the standardised life cycle assessment (LCA) methodology[45,46], we evaluate these impacts in relation to the per capita share of the Earth's carrying capacity using 16 indicators related to climate change, nutrients flows, air pollution, toxicity, and resources use, for which explicit thresholds that should never be exceeded were defined[47–50].

We find that considering current infrastructure, the global average consumption of web surfing, social media, video and music streaming, and video conferencing could account for approximately 40% of the per capita carbon budget consistent with limiting global warming to 1.5 °C, as well as around 55% of the per capita carrying capacity for mineral and metal resources use, 20% for freshwater eutrophication, and over 10% for particulate matter, ecotoxicity, and fossil resources use. A rapid decarbonisation of electricity could reduce the climate change impacts of digital content consumption to just 12% of the per capita carrying capacity by 2030. Yet, concerns surrounding the use of mineral and metal resources may persist, even with extended electronic device lifetimes. Hence, it is essential that roadmaps towards a sustainable ICT sector adopt a more holistic perspective that goes beyond mitigating energy consumption impacts by encompassing measures focused on reducing the extraction of fresh raw materials for electronic devices.

## Results

### Framework for assessing the life cycle impacts of digital content consumption

We cluster digital content consumption into web surfing, social media, video streaming, music streaming, and video conferencing, based on the primary reasons for Internet usage among users between 16 and 64 years[1]. For our analysis, we defined a user archetype representing the global average consumption patterns across all Internet users. We considered, based on the most recent statistics, that the global average user consumes annually 3230 h of digital content, corresponding to 730 h of web surfing, 894 h of social media, 833 h of video streaming (equally shared among standard (720p) and high (1080p) quality), 566 h of music streaming (standard quality), and 207 h of video conferencing (using both audio and video at standard quality). We characterised each digital content with an average data traffic demand; for example, social media consumes an average of 0.31 GB h$^{-1}$, while video streaming consumes between 1.3 and 2.4 GB h$^{-1}$ depending on the streaming resolution. The user accesses digital content through a range of devices, including a smartphone, tablet, laptop, desktop computer, and television. The time dedicated to each digital content is shared among these devices according to recent statistics on users' preferences. For example, social media is accessed through a smartphone 83% of the time, while tablet, laptop, and desktop computer each account for 5.67% of the time (further details, including data sources, can be found in the "Methods" section and Supplementary Section 2).

To quantify environmental impacts, we link the user consumption patterns to the natural resources required and emissions generated throughout the life cycle of the Internet network components, from raw materials extraction to manufacturing, distribution, operation, and end-of-life management. The system boundaries consider data centres, data transmission networks (divided into the access and core networks), customer premise equipment (CPE), and end-user devices (Fig. 1)[10,51]. Data centres and end-user devices process and store data, while the transmission network and CPE (cable modems and Wi-Fi routers) transfer the data between data centres and users[52] (further details of the LCA can be found in the "Methods" section).

## Carbon footprint of digital content consumption

We first assess the carbon footprint of digital content consumption by the global average user, finding that their annual consumption of web surfing, social media, video and music streaming, and video conferencing emits 229 kg CO$_2$-eq year$^{-1}$ (Fig. 2). This value corresponds to approximately 3–4% of the per capita anthropogenic GHG emissions (i.e., 6.0–7.6 tonnes CO$_2$-eq year$^{-1}$ in 2019)[38,53]. However, the impact can vary substantially depending on the carbon intensity of the electricity used to power end-user devices[5,13–15]. Therefore, we extended our assessment to consider the performance of the global average user under the electricity mix of various countries. Here, the impact ranges from 146 kg CO$_2$-eq year$^{-1}$ when considering the Norwegian electricity mix largely based on hydropower (with a carbon intensity of electricity of 0.03 kg CO$_2$-eq kWh$^{-1}$) up to 327 kg CO$_2$-eq year$^{-1}$ for the Indian electricity mix largely based on fossil fuels (with a carbon intensity of electricity of 1.54 kg CO$_2$-eq kWh$^{-1}$).

## Impacts of digital content consumption in relation to the Earth's carrying capacity

We next assess the environmental impacts of digital content consumption against the Earth's carrying capacity considering 16 impact categories for which the maximum impact a natural system could sustain without experiencing irreversible changes was defined[47–50]. Various allocation principles have been proposed to downscale the Earth's carrying capacity to the per capita level, including equality, needs, right to development, sovereignty, and capability principles[42,54–57]. Here, we apply the equality (or equal per capita share) principle which assigns each individual an equal share of the Earth's carrying capacity based on the principles of Sustainable Development that recognise the equal rights to resources of past, current, and future Earth populations[44,58]. Moreover, equality is the most common approach in the literature[55,57], and has been recommended as it works in isolation from complex justice-related and ethical considerations[55].

Under the equality sharing principle, digital content consumption would require on average 41% of the per capita carbon budget consistent with a high likelihood (67%) of limiting global warming to 1.5 °C above the pre-industrial level (Fig. 2). The required share varies from 26% to 61% when considering the respective electricity mix of Norway and India. We here adopt a stringent carbon budget (501 kg CO$_2$ per capita per year, based on the Intergovernmental Panel on Climate Change (IPCC) Sixth Assessment Report (AR6)[59] in line with the precautionary principle followed in the planetary boundaries framework[60,61]. However, the share of digital content consumption varies substantially with the underlying carbon budget, from as high as 55%, for a very high likelihood (83%) of limiting global warming to 1.5 °C, to as low as 7%, for a low probability (17%) of reaching the less ambitious 2.0 °C target (Supplementary Fig. 1).

Beyond climate change, we find that digital content consumption could require on average 55% of the per capita carrying capacity for mineral and metal resources use, 20% for freshwater eutrophication (linked to phosphorus flows), and >10% for marine eutrophication (linked to nitrogen flows), particulate matter, ecotoxicity, and fossil resources use. The remaining impacts are largely below the corresponding limits (<1.5%). The electricity mix used to power end-user devices has little influence on mineral and metal resources use and ecotoxicity as these impacts are mostly dominated by the impact embodied in end-user devices. In contrast, the other most critical impacts can vary considerably with the regional electricity mix. Norway exhibits the lowest impacts across all categories due to the highly renewable electricity mix. Australia and Poland show the highest impact in freshwater eutrophication, primarily due to emissions of phosphorus to groundwater generated by the disposal of lignite mining spoil (lignite accounts for 38% and 29% of the electricity mix in Australia and Poland, respectively). The impact in marine eutrophication is slightly higher in South Africa and India, primarily due to higher

emissions of nitrogen oxides and nitrate from coal-based electricity generation. Moreover, particulate matter formation is more relevant in China due to the higher impact of its coal power plants. Overall, these results suggest that in some impact categories, Internet consumption could leave little room for impacts linked to basic needs like food and transport (recall that all anthropogenic activities should jointly operate within the safe operating space defined by the Earth's carrying capacity), raising concerns about the sustainability level of the current system.

## Contribution analysis of environmental impacts

The breakdown of impacts shows that the impact embodied in the end-user devices (i.e., impacts from raw materials extraction, manufacturing, distribution, and end-of-life management) tends to be the largest contributor towards the total impacts, accounting for an average of 32% of climate impacts, 45% of freshwater eutrophication, 47% of marine eutrophication, 42% of particulate matter, 65% of ecotoxicity, 31% of fossil resources use, 92% of mineral and metal resources use, and 24–76% of the other impacts (Fig. 3). The embodied climate impacts of end-user devices come mainly from electricity consumption for manufacturing, especially for wafers and printed and integrated circuits. Electricity consumption, especially coal-based electricity, is also the major source of the embodied impacts on acidification, photochemical ozone formation, and particulate matter. Conversely, mineral and metal resources use is largely dominated by the mining of gold, a key element used in integrated circuits. Gold mining also emerges as the largest contributor to eutrophication and ecotoxicity impacts –primarily due to the disposal of sulfidic tailings (waste remaining after ore processing)– and land use soil erosion, which is consistent with prior studies that identified gold mining-

related land use as a major source of biodiversity loss[62]. The breakdown of the embodied impacts by end-user device type reveals that the desktop computer is the largest contributor (between 28% and 57% of average embodied impacts, depending on the category, as shown in Supplementary Fig. 2), whereas the impact of the smartphone is the lowest even though it is the most used device (53% of the time spent on digital content).

The impacts attributed to the operation of end-user devices vary depending on the electricity mix at the user location. For example, this stage accounts for an average of 22% of the climate impacts, although it becomes negligible if considering the Norwegian electricity mix (1.3%), and is the largest source of impacts when considering the Indian electricity mix (33%). A similar trend is observed for the other impact categories. The desktop computer is by far the largest contributor to the operational impacts, accounting for an average of 66.2% of the total operational impacts of end-user devices. The operational impacts of the laptop (18.4% of the total), television (8.8%), smartphone (3.7%), and tablet (2.9%) are much lower (Supplementary Fig. 2).

The operation of data centres is responsible on average for 20–30% of climate impacts, acidification, freshwater eutrophication, photochemical ozone formation, particulate matter, ionising radiation, fossil resource use, and land and water use. These impacts are less dependent on the electricity mix at the user location since data centres are spread over the globe. In this regard, the high operational impacts are due to the concentration of data centres in regions where electricity production is currently dominated by fossil fuels, i.e., North America and the Asia Pacific region (39% and 34% of the electricity consumed by global data centres, respectively). The operational and embodied impacts of the CPE, the operational impacts of the data transmission networks, and the embodied impacts of the IT equipment

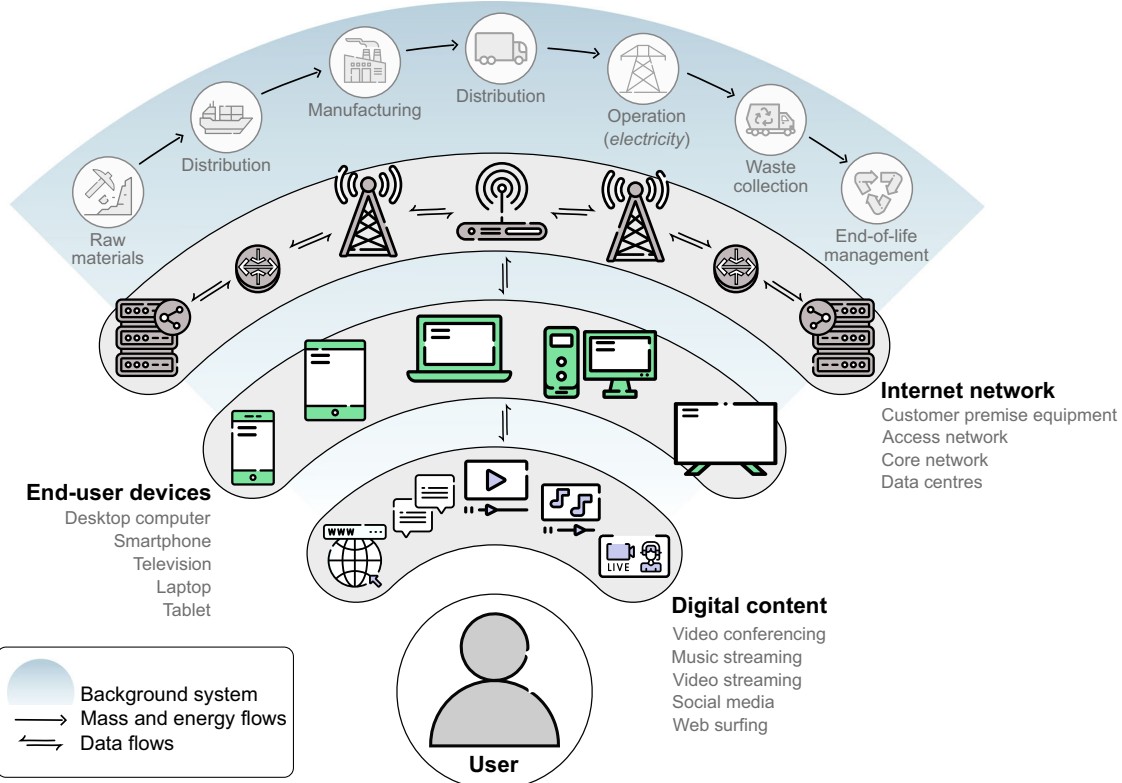

**Fig. 1 | Framework for assessing the life cycle environmental impacts of digital content consumption.** Digital content includes web surfing, social media, video streaming, music streaming, and video conferencing. Data centres and end-user devices process and store data, while the core and access networks and the customer premise equipment (e.g., modems and Wi-Fi routers) transfer the data between data centres and users. The background system supplies the equipment (i.e., end-user devices, modems, servers, etc.) and electricity necessary for operation. The icons used in the figure are designed by Freepik.

used in data centres (i.e., servers and storage equipment) are less relevant.

To shed further light on the impact sources, Supplementary Fig. 3 displays the breakdown of the impacts by digital content. Video streaming generates on average between 40% and 52% of the total impacts, while web surfing, social media, music streaming, and video conferencing contribute each with 10–18%. Notably, video streaming, which is the second most consumed digital content following social media (833 and 894 h year⁻¹, respectively), has the highest data traffic demand, and uses more frequently devices with a high electricity intensity. We note that we considered, based on a survey of global video streaming[63], that TVs, desktop computers, and laptops are used about 53% of the time, while less energy-intensive smartphones and tablets account for 47% (Supplementary Table 3).

## Opportunities for mitigating the impacts of digital content consumption

The results presented above highlight that electricity consumption during the manufacturing and operation of electronic devices emerges

as the primary contributor to the climate and other environmental impacts associated with digital content consumption. Hence, a substantial reduction in GHG emissions of digital content can be anticipated by decarbonising the power sector. To shed light on this issue, we next assess the impacts related to digital content consumption by the global average user while considering alternative electricity generation scenarios by 2030 aligned with three climate targets: a baseline scenario equivalent to a global warming of 3.5 °C and two scenarios aimed at limiting global warming to 2 °C and 1.5 °C, respectively. Internet network-related parameters (e.g., electricity intensity of electronic devices) were kept constant at their current values, implying that any change in environmental impacts can only by attributed to changes in the power sector (additional details can be found in "Methods" section).

In the 3.5 °C scenario, which does not consider any decarbonisation goals, the carbon footprint of the global electricity mix barely changes over time, with only a 3% reduction projected by 2030. Here, digital content consumption would still account for 39% of the per capita carrying capacity for climate change by 2030 (Fig. 4). Other

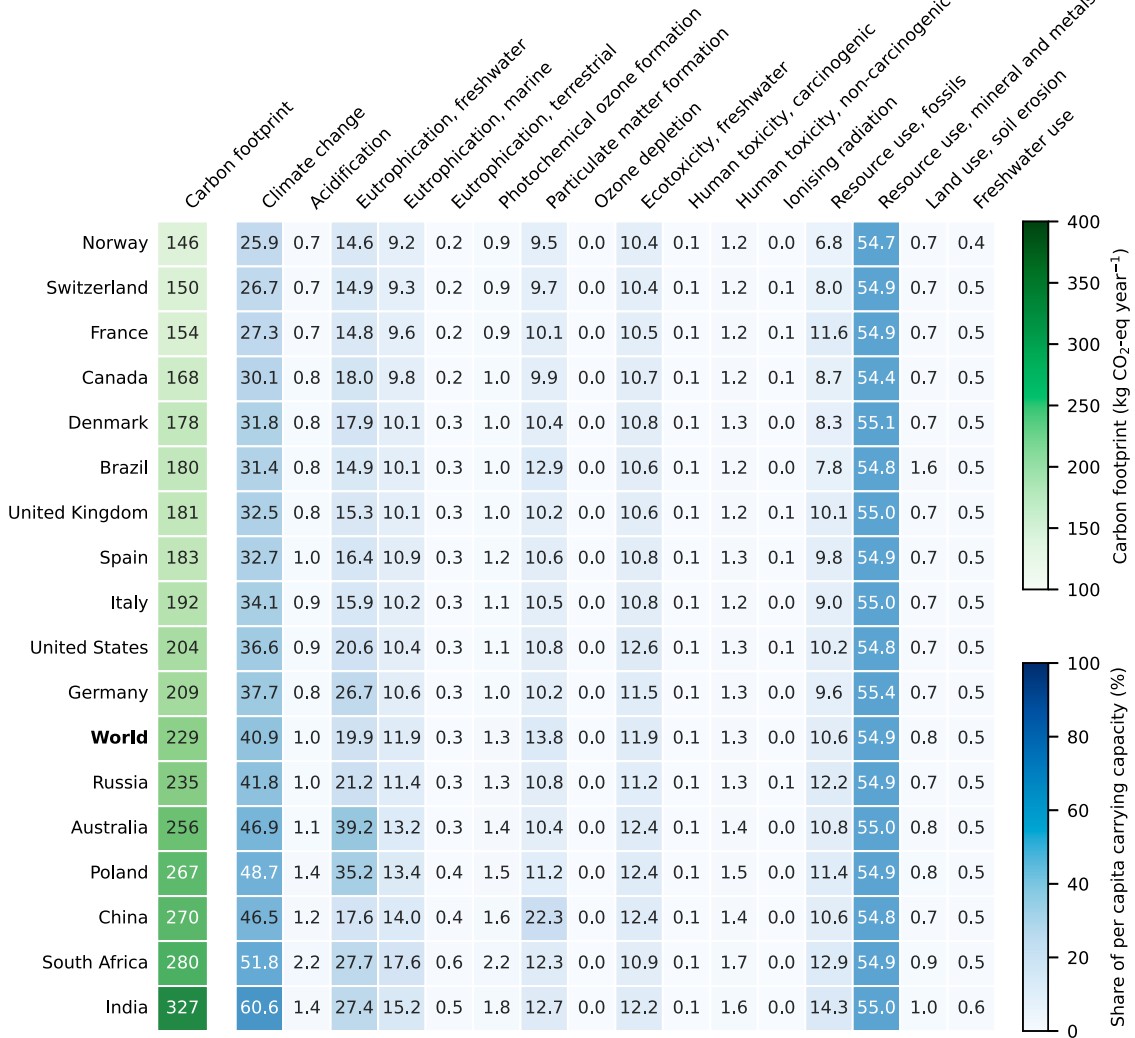

**Fig. 2 | Carbon footprint and share of per capita carrying capacity required by digital content consumption.** Impacts for a user archetype representing the global average consumption patterns across all Internet users. Rows in the heatmap correspond to the electricity mix of various countries ranked according to the carbon footprint (the global average electricity mix is labelled in bold). Very high impacts (global average >40%) are found in climate change and mineral and metal resources use, making it very hard to not transgress the Earth's carrying capacity

considering the remaining goods and services consumed by individuals (i.e., the entire environmental footprint of each individual should not exceed 100%). For more detailed information on the impact assessment methods, their robustness, and the corresponding carrying capacities, see "Methods" section and Supplementary Table 8.

impacts would also remain relatively stable compared to current levels, except for the potential reductions in marine eutrophication (a 21% reduction by 2030) and freshwater ecotoxicity (an 8% reduction by 2030). In the 2 °C scenario, there is a move towards decarbonising the power sector, aiming for a 30% reduction in the carbon footprint by 2030. Nevertheless, this reduction alone would not be sufficient to substantially mitigate the impacts of digital content consumption.

Conversely, the more ambitious 1.5 °C scenario would achieve a 78% reduction in the carbon footprint of the power sector by 2030, primarily due to a large expansion of renewable energies and bioenergy with carbon capture and storage (BECCS). As a result, the climate impacts of digital content consumption could be substantially mitigated due to the decarbonisation of the power sector. Moreover, the decline in fossil fuel usage in this scenario would lead to a gradual decrease in freshwater eutrophication, marine eutrophication, particulate matter, and ecotoxicity, whereas the use of mineral and metal resources would remain almost the same. The decline in the required share of the per capita carrying capacity would be less pronounced due to population growth, i.e., 25% more population by 2050 compared with 2020, thereby reducing the share of the Earth's capacity by 19% (the same annual ecological budget is allocated among a higher global population). When considering a 2030 electricity generation scenario aligned with the 1.5 °C target, digital content consumption would require 12% of the per capita carrying capacity for climate change and less than 10% for ecotoxicity, freshwater and marine eutrophication, and particulate matter. However, it may potentially account for as much as 60% for mineral and metal resources use.

Our findings indicate that a rapid decarbonisation of the power sector could substantially mitigate the climate and other associated impacts linked to digital content consumption. However, concerns would persist regarding the use of mineral and metal resources. It is therefore worthy to implement additional measures aimed at reducing the use of fresh raw materials in electronic devices to achieve a holistic impact reduction. Extending the lifetime of electronic devices can play a prominent role in this endeavour. For example, doubling the lifespan of electronic devices holds the potential to diminish mineral and metal resources use from 55% of the per capita carrying capacity to 29%, considering the current electricity generation scenario, or from 60% to 32% when considering the 1.5 °C-aligned 2030 scenario (Fig. 5). Lifetime extension affects the other impact categories as well, as the embodied impacts of electronic devices would be distributed over a higher number of operational years. Nonetheless, the mitigation potential for other impacts like freshwater and marine eutrophication, particulate matter, or ecotoxicity remains limited.

## Discussion

Assessing and monitoring the environmental impacts of consumption is crucial for the attainment of sustainable development goal 12 (SDG 12) on responsible production and consumption. Our work further expands current efforts by proposing a bottom-up framework to assess the impacts of Internet consumption considering all the necessary infrastructure linked to the users' consumption patterns.

The results reveal that the global average digital content consumption considering current infrastructure would require >40% of

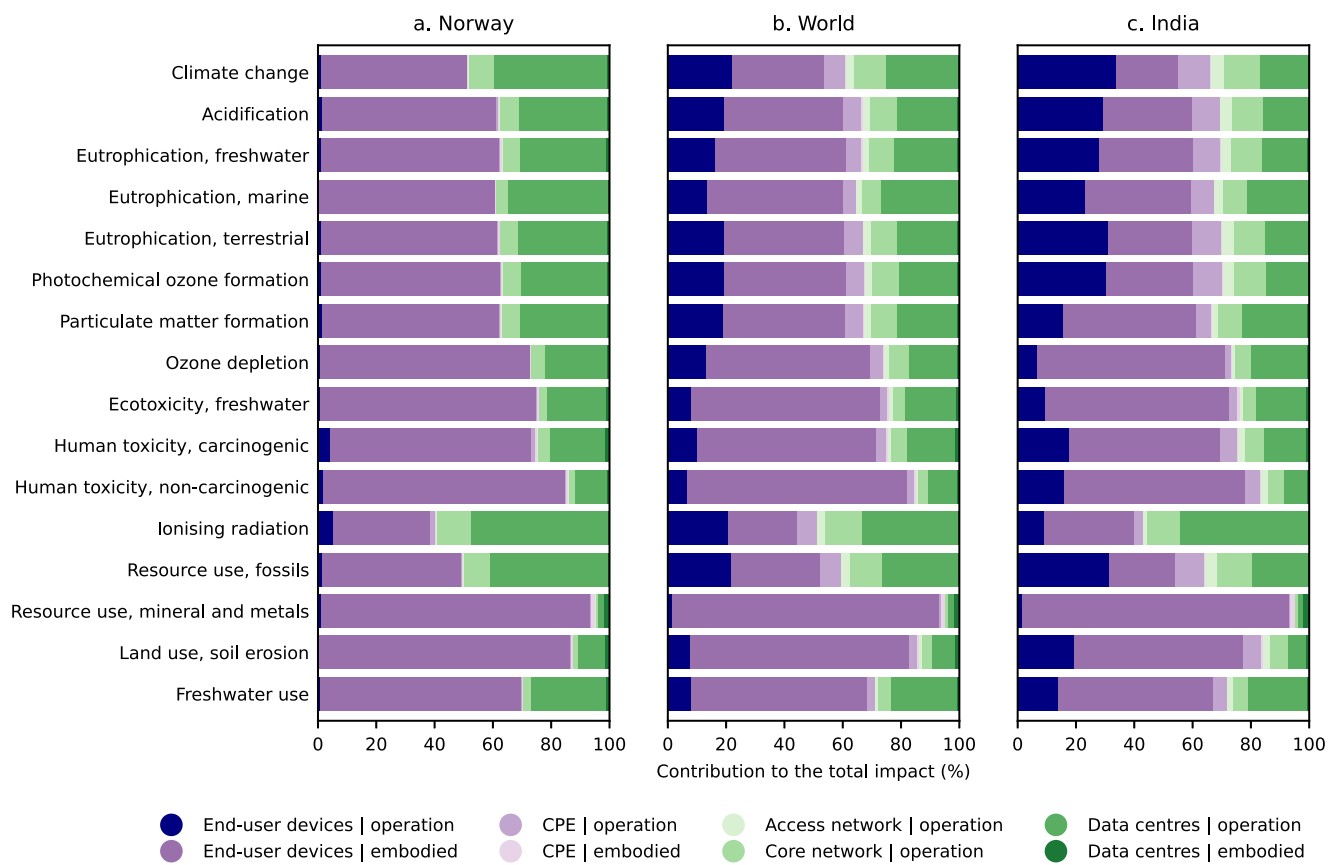

**Fig. 3 | Contribution of Internet network components to the life cycle environmental impacts of digital content consumption under three electricity mixes used to power end-user devices. a** Norwegian electricity mix, **b** global average electricity mix, and **c** Indian electricity mix. These mixes were chosen to represent a very low (Norway), average (World), and very high (India) GHG emission intensity of electricity. Operation refers to impacts from the consumption of electricity in the use stage, while embodied refers to impacts from raw materials extraction, manufacturing, distribution, and end-of-life management. Overall, the impact embodied in the end-user devices generally dominates the total impacts. CPE: customer premise equipment.

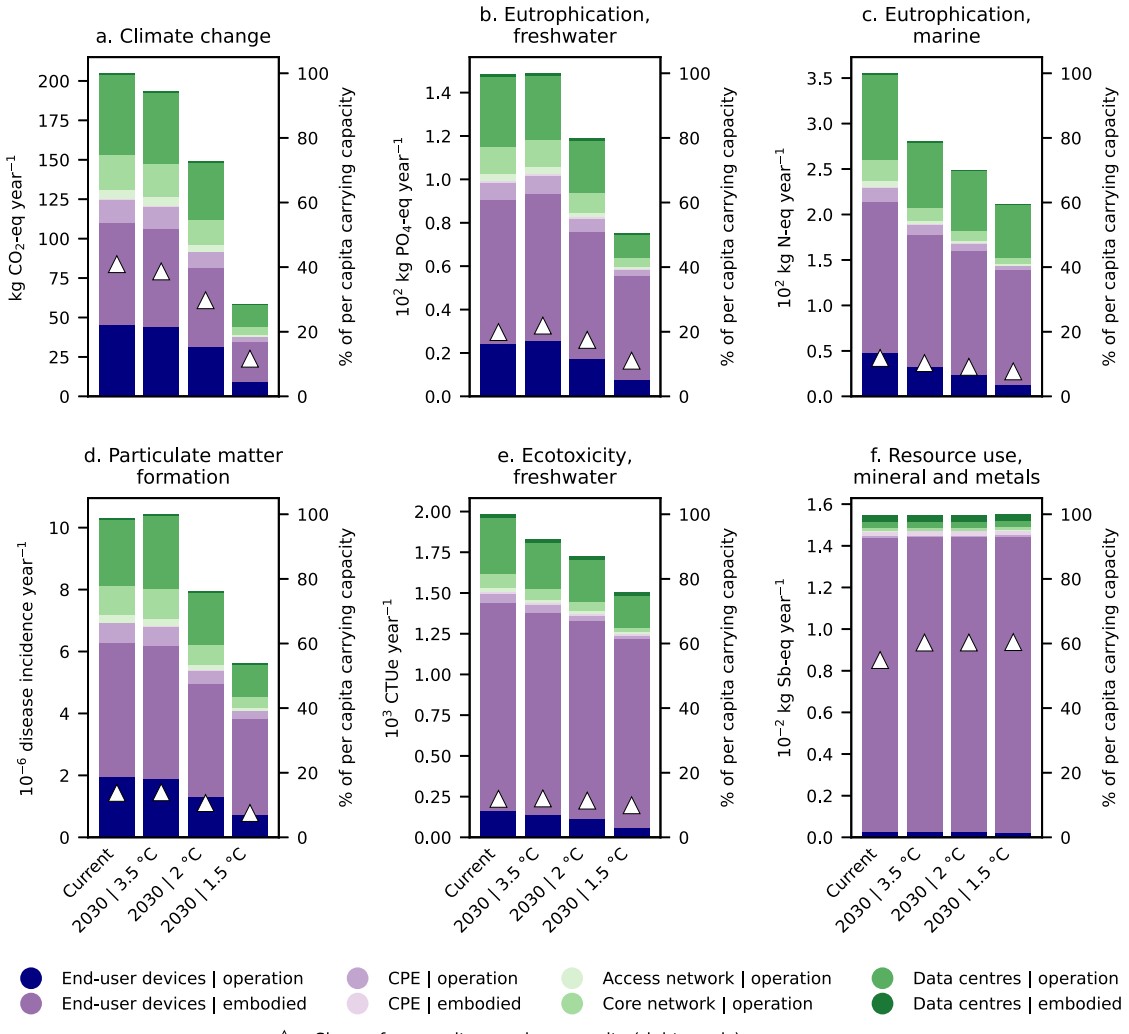

**Fig. 4 | Life cycle environmental impacts and share of per capita carrying capacity required by digital content consumption considering 2030 electricity generation scenarios compatible with limiting global warming to 3.5 °C, 2 °C, and 1.5 °C.** Impacts for a user archetype representing the global average consumption patterns across all Internet users. Only the most critical impact categories are displayed: **a** climate change, **b** freshwater eutrophication, **c** marine eutrophication, **d** particulate matter formation, **e** freshwater ecotoxicity, and **f** mineral and metal resources use (fossil resources use is omitted since it is strongly connected with the climate change category). For more detailed information on the impact assessment methods used and their robustness and the corresponding carrying capacities, see "Methods" section and Supplementary Table 8. CPE customer premise equipment.

the per capita Earth's carrying capacity for climate change and mineral and metal resources use, >20% for freshwater eutrophication, and >10% for marine eutrophication, particulate matter, ecotoxicity, and fossil resources use. Notably, an average world citizen already exceeds the limits for climate change and particulate matter, while freshwater eutrophication, fossil resources use, and mineral and metal resources use are within the zone of uncertainty[47,61]. The food system and transport are typically identified as priority areas to reduce these impacts within the safe zone[41,64–67], but our findings highlight the importance of not overlooking efforts to mitigate the impacts of the ICT infrastructure to avoid exacerbating further the pressure on the finite Earth's carrying capacity.

We emphasise the crucial role of decarbonising electricity production in mitigating the climate impacts of digital content consumption. ICT companies are increasingly concerned about the environment and aim to become carbon neutral or even negative. Sustainability commitments primarily focus on renewable energies, with increased corporate momentum on carbon dioxide removal (CDR)[68]. However, the ICT infrastructure entails environmental impacts associated with the extraction and processing of raw materials used in electronic devices that cannot be mitigated by deploying renewable energies or CDR. Additionally, these impacts have a strong regional component, such as those related to nutrients emissions, ecotoxicity impacts, and land use, and typically occur in geographical locations distinct from where digital content consumption takes place. Besides improving the mining and metal processing industry[69], lying beyond the scope of our study, here we demonstrate the potential environmental benefits of extending the devices lifetime, which could be appealing to both businesses and consumers. Business models in the ICT sector are transitioning from selling a piece of equipment to providing a service for several years[70]. Consequently, there is an economic interest in increasing the lifetime of electronic devices. Extended producer responsibility, which holds producers responsible for the entire lifetime of their products, may further encourage the durability of electronic devices[71]. Furthermore, the transition towards a more circular economy could play a pivotal role in reducing the environmental impact of digital content. An increased reutilisation of electronic devices, e.g., via the second-hand market, can contribute

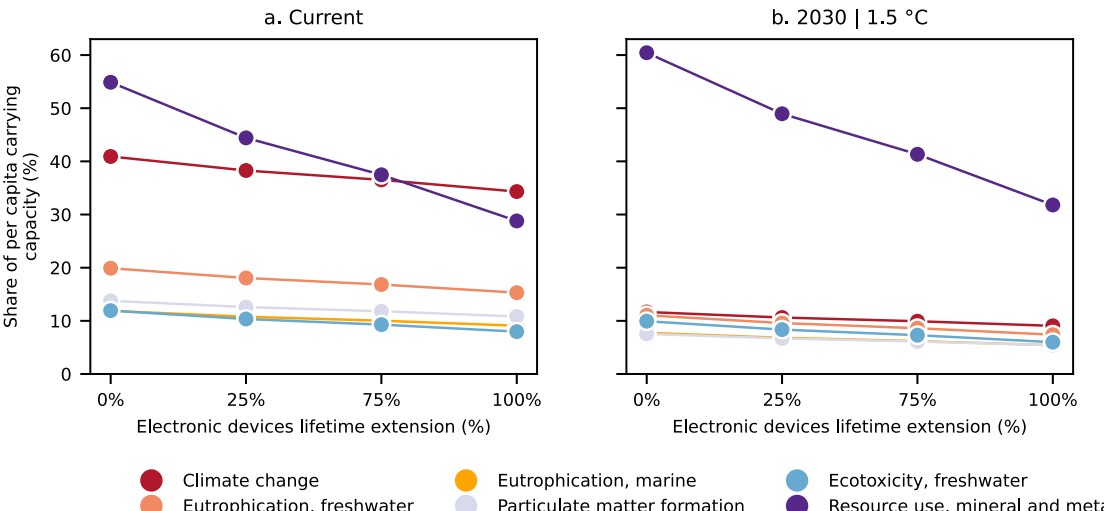

**Fig. 5 | Share of per capita carrying capacity required by digital content consumption over a year vs. electronic devices lifetime extension.** Impacts for a user archetype representing the global average consumption patterns across all Internet users considering **a** the current electricity mix scenario and **b** a 2030 electricity generation scenario compatible with limiting global warming to 1.5 °C. Only the most critical impact categories are displayed (fossil resources use is omitted since it is strongly connected with the climate change category). For more detailed information on the impact assessment methods used and their robustness and the corresponding carrying capacities, see "Methods" section and Supplementary Table 8.

notably to lifetime extension and embodied impacts mitigation. Yet, the materialisation of reuse strategies will ultimately depend on consumer preferences and price attractiveness over a new device[72].

The climate impacts of digital content consumption exhibit substantial variability across countries, primarily driven by differences in the carbon intensity of electricity generation. Our findings indicate that digital content consumption may impose a relatively modest burden on the per capita carbon budget in countries with a low-carbon electricity mix, such as Norway, whereas it can constitute a large portion of the entire budget in countries heavily reliant on fossil fuels, such as India. The interpretation of these results can indeed raise potentially problematic implications, suggesting that users in regions with a carbon-intensive electricity mix should consider reducing their consumption to align with sustainability targets. However, a more nuanced understanding reveals the importance of context-specific mitigation strategies. While it remains imperative for both the energy and ICT sectors to reduce emissions worldwide, countries where the use phase has a more modest environmental impact due to a cleaner electricity mix should focus on strategies aimed at diminishing embodied impacts, including extending the lifetime of electronic devices and recycling. Conversely, in regions with carbon-intensive electricity mixes, the greatest potential for mitigation lies in the urgent decarbonisation of their electricity production. This outcome carries major implications in the context of the anticipated widespread adoption of the Internet by a substantial portion of the approximately 3 billion unconnected people in the coming decades. Notably, the majority of these unconnected individuals reside in fossil fuels-reliant countries such as India, China, Pakistan, and Nigeria[1]. Finally, it is equally important to acknowledge the validity of our findings within the context of the equality principle employed for downscaling the Earth's carrying capacity. Utilising alternative principles that rely on differentiated responsibilities would result in higher carbon budgets allocated to users in developing countries, potentially diminishing the importance of digital content consumption from an environmental viewpoint while amplifying its relevance for users in developed countries. However, it is worth noting that a broadly accepted principle beyond the equal per capita share is currently lacking[58]. In particular, choosing an alternative principle introduces a level of

subjectivity regarding which approach may be considered more ethically appealing[44].

Our results are subject to the inherent uncertainties in LCA studies concerning life cycle inventory (LCI) data, impact assessment methods, and methodological choices (Supplementary Section 4)[73,74]. We assessed the robustness of our conclusions under uncertainties in the LCI data (i.e., electricity intensity of electronic devices, equipment requirements, manufacturing processes, power generation, etc.) via the Monte Carlo sampling method (Supplementary Fig. 8). For example, we find that the uncertainty range of the required share of the per capita carrying capacity is 40–59% for climate change, 46–97% for mineral and metal resources use, and 15–63% for freshwater eutrophication (based on a 95% uncertainty range). Therefore, based on these ranges, LCI uncertainties might not change the general outcome of our analysis. Regarding the impact assessment methods, we used the most up-to-date and comprehensive methods as recommended by the IPCC[59] and European Commission[75]. Yet, the robustness of the underlying method varies across impact categories. The assessment methods for climate change and particulate matter formation are recommended and satisfactory, while the methods for the other critical impacts are recommended but either need further improvements (freshwater eutrophication and ecotoxicity) or should be applied with caution (mineral and metal resources use and land use)[76]. Finally, in our analysis, we have considered a user archetype that represents the global average consumption patterns across all Internet users, thus overlooking the large heterogeneity in consumption levels across countries. This limitation stems from data gaps that prevented us from defining user archetypes specific to individual countries. However, it is important to acknowledge that this assumption might lead to an overestimation of the impacts in countries where average consumption levels are lower. To shed further light on the implications of this approximation, we have considered two hypothetical users representing two extreme consumption patterns, finding that the most critical impacts could decrease by 53–67% or increase by 21–31% relative to the average user (see Supplementary Figs. 6 and 7). Improving the availability of data on user consumption patterns would enable more comprehensive studies that consider the geographical variation of consumption patterns. Despite the aforementioned limitations, we believe that the methodological framework and data used allow us to

generate valuable insights into the so far poorly understood environmental footprint of digital content consumption.

## Methods

### Methodology overview

LCA is the predominant methodology to assess the potential environmental impacts over the life cycle of products and services, from raw materials extraction through production, distribution, use, and end-of-life disposal[77]. LCA provides a systematic and holistic way to compare different consumption options[41,78,79]. However, LCA alone provides limited insights into whether these options enable an environmentally sustainable lifestyle. To overcome this limitation, LCA studies have begun to incorporate the Earth's carrying capacity as an environmental sustainability reference against which anthropogenic impacts should be assessed[47–49]. In this context, the carrying capacity is the maximum impact a natural system can tolerate without undergoing irreversible negative alternations in structure or function[48]. Here, we combine LCA and the Earth's carrying capacity framework to shed light on the environmental footprint of digital content consumption. The LCA is conducted in accordance with the ISO 14040/14044 standards, including the four phases described in detail next[45,46].

### Goal and scope definition

We conduct an attributional LCA to quantify the environmental impacts of digital content consumption and to compare these impacts against the per capita share of the Earth's carrying capacity. A consumption-based perspective is adopted, meaning that the impacts generated throughout the life cycle of Internet connection delivery are assigned to the user who consumes digital services. The functional unit is defined as the digital content consumption of an average Internet user over a year, equivalent to 3230 h of digital content divided into 730 h of web surfing, 894 h of social media, 833 h of video streaming, 566 h of music streaming, and 207 h of video conferencing (Supplementary Table 2). Additional activities and services dependent on the Internet, such as Internet of Things (IoT) devices or video gaming, have been omitted in this study. Their inclusion would amplify the overall impacts making them more severe than what was considered herein. Therefore, we report lower bounds on the "true" impacts that could be determined if all data on digital consumption were available. We used recent statistics on users' preferences to define the devices (i.e., smartphone, laptop, television, etc.) and settings (e.g., video streaming resolution) used to access digital content (Supplementary Table 3). The consumption behaviour and user's preferences are based on the latest available data for the period 2019–2022, while the reference year for technology data (e.g., electricity intensity of electronic devices) corresponds to 2020, whenever possible. In cases where technology data was not available for 2020, we extrapolated the most recent available data to 2020, e.g., by considering an annual energy usage improvement.

Given the high complexity of the Internet, the definition of the system boundaries is crucial. For example, discrepancies in the definition of system boundaries have resulted in large variations in the electricity intensity of the Internet[11]. Here, we adopt a cradle-to-grave scope considering raw materials extraction, manufacturing, distribution, operation, and end-of-life management of all the major components of the Internet network, including data centres, data transmission networks, CPE, and end-user devices (Fig. 1).

### Life cycle inventory

In the inventory analysis phase, all the relevant input and output flows associated with digital content consumption are quantified (i.e., energy, raw materials, emissions, etc.). We developed tailored inventories to represent the patterns of digital content consumption as well as the operation of the Internet network components, while the LCIs of the background processes (e.g., electricity generation and electronic devices manufacturing) were retrieved from the ecoinvent database v3.8 (cut-off system model)[80].

As noted by Billstein et al.[81], there is currently no standardised way of measuring Internet electricity usage. Following their recommendations, we consider that the electricity intensity of data centres and the core network is proportional to the load and, consequently, is expressed as the amount of electricity required per unit of data traffic (i.e., kWh GB$^{-1}$). In contrast, end-user devices, CPE, and the access network are considered agnostic to data load and their electricity intensity is expressed as the amount of electricity required per active hour (Supplementary Table 4). The electricity consumed by the operation of end-user devices, CPE, and the access network is supplied by the national electricity mix according to the user location. Since data centres are distributed globally, we consider the regional distribution based on Masanet et al.[6]. Consequently, the electricity required by data centres is supplied by regional electricity mixes in North America (39%), Asia (34%), Europe (22%), Latin America and the Caribbean (3%), and Middle East and Africa (2%). While an increased procurement of renewable electricity by data centres has the potential to mitigate certain impacts, particularly climate change, its overall implications on our conclusions remain marginal (refer to the sensitivity analysis in Supplementary Fig. 4). Moreover, as the core network serves as the backbone connecting data centres and users, its electricity requirements are distributed among user and data centres locations. Due to the lack of more specific data, we have assumed an even distribution between the two locations, in line with the findings presented by Coroama et al.[12].

The impact embodied in end-user devices and CPE were normalised per active hour considering the total amount of active hours over their operating lifetime (Supplementary Table 5). On the other hand, the impact embodied in data centres IT equipment (i.e., servers and storage equipment) were normalised per GB of data traffic based on the global stock of servers and storage capacity in data centres, the outbound data traffic, and an average equipment lifetime (Supplementary Table 6). The equipment required for the access and core networks have been omitted due to the lack of data and the substantially higher environmental importance of the use stage[15,17,27]. Moreover, note that the inventories for the manufacturing of the electronic devices (including the extraction of raw materials, distribution, and end-of-life disposal) were retrieved from the ecoinvent database.

Operation- and infrastructure-related inventory data are linked to digital content consumption patterns by explicitly including the data traffic demand of each digital content (Supplementary Table 7) as well as the time dedicated to each digital content through each device (see "Goal and scope definition" section). Such a linkage enables our model to evaluate changes in environmental impacts as a consequence of behavioural changes.

For the prospective analysis, we used the open-source tool premise v1.4.2[82] to systematically change the LCIs in the background ecoinvent database based on the output scenario results from the Integrated Assessment Model (IAM) IMAGE[83]. In essence, premise modifies the LCIs for the electricity, steel, cement, and transport sectors to reflect the evolution over time as described by the three climate scenarios (i.e., 1.5 °C, 2 °C, and baseline (3.5 °C) scenarios). The three scenarios assume a "Middle of the Road" socioeconomic pathway (SSP2), meaning that economic and societal development are in line with historical trends.

Calculations were performed with Brightway2 (v2.4.4), a Python-based open-source LCA software[84]. The code to import the LCIs and perform the LCA calculations can be found in ref. 85, which ensures reproducibility and will allow other researchers to define new user archetypes and perform further studies.

## Life cycle impact assessment

In this phase, the LCI elementary flows (i.e., emissions and natural resources) are translated into potential impacts by using a set of characterisation factors. We evaluate the 16 impact categories included in the Environmental Footprint (EF) methods recommended by the European Commission[75]. The impact assessment is performed with the methods recommended in EF 3.0[75] with a few exceptions. Specifically, we apply the IPCC 2021 Global Warming Potentials (GWP) for a 100-year time horizon[59] to quantify the carbon footprint, use the updated methods available in EF 3.1[86] for ecotoxicity and human toxicity assessment, employ the updated LANCA model[87] for land use impact assessment, and apply the water withdrawal flows to quantify freshwater use. Further details on impact categories and assessment methods can be found in Supplementary Table 8.

## Interpretation

As mentioned previously, standard LCAs lack a reference against which anthropogenic impacts should be assessed to determine whether they exceed sustainable levels[88]. Here, we compare the impacts associated with digital content consumption to the per capita share of the Earth's carrying capacity for each impact category. We define the carrying capacity for climate change based on the Paris Agreement goal of limiting global warming to well below 2 °C, preferably to 1.5 °C, above pre-industrial levels. Following the precautionary principle, we use the IPCC AR6 estimates of the remaining carbon budget (i.e., remaining $CO_2$ emissions) from the beginning of 2020 until global net zero $CO_2$ emissions are reached with a high probability (67%) of limiting global warming to 1.5 °C compared to pre-industrial levels[59]. For other impact categories, the carrying capacity is defined based on the maximum allowable impact derived from science-based thresholds[47–50]. The approach and assumptions for determining the Earth's carrying capacity are documented in Supplementary Section 3.

Since carrying capacities are derived from global sustainability thresholds, they need to be distributed among the global population. In this study, we apply the equality principles, assigning an equal share per capita following the approach described in Dao et al.[58]. Accordingly, the calculation rule differs depending on whether the carrying capacity is a cumulative budget (climate change) or an annual budget (the other impact categories). For climate change, the per capita carbon budget is calculated as the ratio between the cumulative carbon budget over the period 2020–2100 and the cumulative global population over the same period (based on world population prospects from the United Nations[89]). For the other impact categories, the per capita carrying capacity is computed as the ratio between the annual carrying capacity and global population in the reference year (i.e., 2020 or 2030).

## Reporting summary

Further information on research design is available in the Nature Portfolio Reporting Summary linked to this article.

## Data availability

The raw data used to create the LCIs are provided in the Supplementary Information file. The LCI datasets generated in this study are available at https://doi.org/10.5281/zenodo.8122381. The background LCI datasets used in this study are available in the ecoinvent v3.8 (cut-off system model) database under the accessible link https://ecoinvent.org/.

## Code availability

The complete Python code used for the calculation and visualisation of the results is available at https://doi.org/10.5281/zenodo.8122381.

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

## Acknowledgements

This publication was created as part of NCCR Catalysis (grant number 180544), a National Centre of Competence in Research funded by the Swiss National Science Foundation. R.I., V.T., R.N.G., W.J.S., and G.G.-G are all authors affiliated with NCCR Catalysis.

## Author contributions

R.I., G.G.-G., V.T., R.N.G., and W.J.S. designed the study. R.I. collected the data and conducted the life cycle assessment. R.I. and G.G.-G. wrote the paper. R.I. and V.T. designed the figures. R.I., V.T., R.N.G., L.V., W.J.S., and G.G.-G. interpreted the results and reviewed.

## Competing interests

The authors declare no competing interests.
