## [Peer Review File · Nature Communications]

The environmental sustainability of digital content consumptionReviewers' Comments:

Reviewer #1:

Remarks to the Author:

The article is a valuable contribution, which presents the relevance of digital consumption for the achievement of environmental protection goals. What is particularly original is that the authors put the footprint in relation to the requirements for lifestyles that are in line with long-term environmental protection goals. Certainly, there are other areas that are more acute for environmental protection (e.g. food, transport, housing), and yet this paper shows impressively that the digital sector also has a contribution to make.

Still, I believe there is some room for improving the article further, which I list below.

Please note, that I can review climate-related impacts of digital technologies, and the results provided by the authors for this impact category seem plausible to me. However, I am not an expert in other impact categories (e.g. particulate matter, freshwater eutrophication) and cannot assess the plausibility of these results.

1. The abstract and the results section is missing reference years, e.g. "finding that web surfing, social media, video and music streaming, and video conferencing could consume on average ~40% of the per capita carbon budget consistent with limiting global warming to 1.5 °C,..."
2. Also, you could make it more clear that your study takes a global perspective. E.g. in row 60 you talk about the "studied user". Is it the global average across all Internet users?
3. I would briefly mention in row 41+ that you conducted a LCA, and which standards you applied.
4. In the abstract, you mention that behavioral change is required. This is very broad and should be specified.
5. You mention that consumers spend daily seven hours on the Internet. I am wondering if this is only active time spent on the Internet or also passive time spent on the Internet, e.g. streaming music in the background. I suppose, many devices are exchanging data with the "Internet" in the background, even though users conduct other activities. This could be specified.
6. You state that only "recently" the environmental footprint of digital ICT gained interest. This might be true for the "public debate". However, ICT companies (such as telecommunication companies) and academia deal with this issue since at least 20 years. For example, see the studies by the Global e-Sustainability Initiative, Sustainability Reports by telcos such as Deutsche Telekom, or work published in the scope of the ICT for Sustainability or Environmental Informatics conference series.
7. You state that the ICT sector causes roughly 1.2% of global anthropogenic GHG emissions. Recent meta-studies provide a different range, e.g.
<https://www.sciencedirect.com/science/article/pii/S2666389921001884>
<https://www.sciencedirect.com/science/article/pii/S0195925522002992>
8. In the introduction, you mention the results of your study already. From my perspective, these should only be mentioned in the abstract or results section, see row 45+
9. "Norway exhibits the lowest impacts due to its highly renewable electricity mix, while the highest impacts are found in Australia (freshwater eutrophication), South Africa (marine eutrophication), China (particulate matter), and India (fossil resources use).  To me, it is unclear how the electricity mix relates to freshwater eutrophication, marine eutrophication, PM,....
10. In Figure 3 you differentiate between operation and embodied impacts for all elements but the access and core network. To me, it is unclear why you do so and what life cycle phases have been considered in the access and core network. This also applies to Figure 4.
11. In Figure 4a, the black arrow is unclear to me.
12. Potentially, the readability of the results section could benefit from some subheadings.
13. In line 167+ you describe how you modeled the future scenario in regard to climate impacts. In line 174 you suddenly switch to other impact categories (e.g. freshwater eutrophication,...) without explaining how you modeled future scenarios for these. I assume these are not only related to the future climate scenarios.

14. There are some grammatical/language errors in the paragraph in line 209+

15. Did you consider that demand for digital services could not only rise because of population growth and increasing access to digital services, but also due to rebound effects. As digital services become cheaper or more performant, demand per capita might also increase (e.g. cheaper access to faster broadband speeds often lead to higher demand for data traffic). This has been addressed in this paper with regard to 5G networks: https://link.springer.com/chapter/10.1007/978-3-031-18311-9_13
It is probably difficult to model these effects for the future, but they could be at least discussed.

16. I am aware that you did an uncertainty analysis with a Monte Carlo Simulation. And you state that your results do not change significantly, even if you consider the uncertainty. I am quite surprised that this is the case, given all the uncertainty regarding life cycle inventory data of digital devices and infrastructures. There is large uncertainty regarding production impacts of end-user devices (which you addressed) but also regarding the general "energy intensity of the Internet". For example, the study by the French Think Tank the Shift Project, was criticized for pessimistic assumptions regarding energy intensity (

<https://www.iea.org/commentaries/the-carbon-footprint-of-streaming-video-fact-checking-the-headlines>). The references 10 and 11 in the you use in the supplementary material are already quite old, even though I am not certain if more recent studies exist.

Even if you considered all sources of uncertainty, I think you should mention your results for a worst case (most pessimistic assumptions) and best case (most optimistic assumptions), as well as limitations in the availability of up-to-date data (if this is true), in the main text.

17. You use different terms without explicitly stating what is meant by it, in particular the ICT sector, the Internet. If I understand correctly, you included end user devices, data centers and networks in your calculation. The aggregation of these components is usually called the ICT sector. However, not all data centers or network (usage) are part of the Internet, because the Internet is "just" the part of the infrastructure which is used for certain protocols (e.g. WWW). For example, some infrastructure (e.g. corporate data centers and networks) are used for other applications beyond "the Internet". I recommend clearly stating the system boundary and avoiding confusion between the two terms early in the article.

18. In my understanding, you used the regional average electricity mixes. Some ICT companies are more advanced in their use of electricity from renewable energies than the regional average. You could mention that in line 322.

Reviewer #2:

Remarks to the Author:

REVIEW OF "THE ROLE OF DIGITAL CONTENT CONSUMPTION IN ENVIRONMENTALLY SUSTAINABLE LIFESTYLES"

While this article tackles an important topic and it comes from a credible research team, it falls prey to several common errors and conveys illusory precision about the next few decades of information and communication technology (ICT) emissions. I recommend it be rejected in its current form.

It is a mistake to project emissions of computing technologies more than a few years ahead. Beyond typical manufacturer product roadmaps (which typically extend for 3-5 years), the characteristics of ICT technologies are impossible to predict. This article attempts to predict to 2050, which is absurd for ICT.

This article also cites high-end estimates of ICT electricity use that are not credible, without explaining that context. For example, the work of Anders Andrae consistently overstates projected ICT electricity use, and when those growth rates don't materialize, Andrae just starts at the latest estimates of current electricity use (which is 3-5% for all ICT) and then projects rapid growth again, which always fails to materialize.

There is a tendency to overestimate electricity use and emissions of ICT that is well documented [1]. One reason for that tendency is that forecasters ignore or underestimate the potential for efficiency improvements to offset most or all of growth in ICT service demand. As an example, data center services from 2010 to 2018 grew very rapidly (with compute instances, data flows, and data storage capacity growing manyfold), but total electricity use for data centers only grew 6% over that period [2]. Only in special cases like 2000 to 2005, when data center infrastructure was being built out for the first time in the US and Europe, does the data show very rapid growth in total electricity use by data centers [3].

Estimates of life cycle emissions for manufacturing of products always lags a lot, and so it's difficult to take seriously estimates from even a few years ago (because they are based on data from a few years before that), and things change so fast in the technology industry.

Understanding energy, emissions and materials impacts from ICT is important, but projecting to 2050 makes no sense. I suggest a more modest goal of accurately tallying current ICT impacts (including emissions, energy, and materials) for as late a year as the data support. That would be a contribution to the literature, as long as the caveats about the inadequacies of underlying data (particularly for embedded emissions and material use) are sufficiently well explained. That is a major reframing of this work, but if the authors are willing to undertake it, I would be willing to review it again.

REFERENCES

1. Koomey, Jonathan, and Eric Masanet. 2021. "Does not compute: Avoiding pitfalls assessing the Internet's energy and carbon impacts." *Joule*. vol. 5, no. 7. June 24. pp. 1625-1628. [<https://www.sciencedirect.com/science/article/abs/pii/S2542435121002117>]
2. Masanet, Eric, Arman Shehabi, Nuo Lei, Sarah Smith, and Jonathan Koomey. 2020. "Recalibrating global data center energy-use estimates." *Science*. vol. 367, no. 6481. Feb 28. pp. 984. [<http://science.sciencemag.org/content/367/6481/984.abstract>]
3. Koomey, Jonathan. 2008. "Worldwide electricity used in data centers." *Environmental Research Letters*. vol. 3, no. 034008. September 23. [<http://stacks.iop.org/1748-9326/3/034008>]

Reviewer #3:

Remarks to the Author:

This paper investigates the anthropogenic impact of users' digital content consumption across various environmental factors including climate, water, eutrophication, mineral and material use, etc. The authors consider the end-to-end impact of computing devices from raw material procurement to hardware manufacturing to operational use. A number of devices are considering including consumer electronics (e.g., smartphones, tablets, laptops), networking systems, and data centers.

The results of the analysis demonstrate a few noteworthy results:

The environmental impact of computing platforms owes not only to operational use but also raw material procurement and hardware manufacturing. In fact, in many cases, embodied environmental impacts outweigh the operational environmental impacts.

While ICT emissions currently account for a small portion of climate impact, if left unchecked, ICT will account for a large portion of per capita carrying capacity, in order to limit the global warming to within 1.5 degrees celsius in the coming decades.

A collection of efficiency improvements, lifetime extension, recycling, and behavioral change is needed to limit the climate impact of ICT.

Overall, I found the paper to be a very interesting read. While I have seen articles in the past that

mention the embodied carbon of ICT emissions [1, 2] from the computer systems community, I have not seen the climate impact being put into context of the per capita carbon capacity of individuals and the world. This is a powerful result that can help encourage future research and investigation around sustainable computing. I also found the analysis going beyond carbon to include acidification, freshwater, eutrophication, particulate matter formation, fossil resource use, and land and water use to be helpful to understand a more holistic environmental footprint of ICT.

In terms of the methodology, the methodology adopted by the authors appears sound. There are a few assumptions and parameters that could be further substantiated or extended to explore additional scenarios (see below for questions) but overall the methodology seems appropriate. The use of life cycle analysis follows from established practices in the community and the comparison to per capita climate carrying capacity is based on targets from standardized communities and organizations. The supplemental information provides sufficient information to reproduce the results provided in the paper.

There are a couple of specific questions I have for the authors:

In the introduction, you mention "data centers and data transmission networks jointly account for 2-3% of global electricity consumption" while "ICT sector [...] emitted 700 Mt CO₂-eq 2020 (equivalent to 1.2% of global anthropogenic greenhouse emission)". Could you expand on the gap between 1.2% of GHG emissions from ICT in total and 2-3% of electricity consumed by data centers alone?

Furthermore, does the 1.2% of GHG emissions also include emissions from hardware manufacturing?

On page 4 you take the global average electricity mix (0.68 kg CO₂-eq per kWh). However, as you mention in your methodology, this does not take into account the distribution of users and use patterns across the world (for instance, you say the remaining 40% of unconnected users in the world come largely from India and China which have higher carbon intensity than the world average). How sensitive are your results to the electricity mix and use patterns across the world?

On page 8 you mention "the embodied impacts of the IT equipment used in data centers (e.g., servers and storage equipment) are less relevant". Why is this? In fact, for many hyperscale data centers (e.g., Google, Meta, Microsoft, Amazon) that are procuring an increasing amount of renewable energy to power their operation, the majority of their environmental impact owes to embodied emissions [1, 2]. Can this impact be taken into account? Or, should the reader see the estimates as a lower bound for ICT's impact.

Following from the previous point Figure 3 considers operational and embodied emissions for all components except the access and core networks. Why are these not considered? Or is there insufficient data to estimate the embodied impacts of the networking systems?

It is unclear in Figure 4 if "climate impact" (top left plot) is solely from digital content consumption (the operational emissions from running devices) or also from manufacturing the devices used. The analysis also assumes a high degree of carbon capture technology which has not been demonstrated to be effective at a large scale. Would it be possible to have two scenarios, with an optimistic and more pessimistic view of renewable energy/carbon capture technology? Without this context, I wonder if the conclusion is "efficiency" and carbon-optimization for ICT is not needed given renewable energy and carbon capture will eliminate the climate impact of ICT devices.

Once again, overall I feel the paper makes strong contributions to understand the holistic environmental impact of technology and motivates future work. I hope the authors can address the questions above to clarify some of the details around the work!

[1] Chasing Carbon: The Elusive Environmental Footprint of Computing

[2] The Dirty Secret of SSD's: Embodied Carbon

[3] GreenChip: A tool for evaluating holistic sustainability of modern computing systems

Reviewer #4:

Remarks to the Author:

Dear author(s) and/or editor(s),

I would like to thank you for the opportunity to review your manuscript "The role of digital content consumption in environmentally sustainable lifestyles". I have now completed my review and would like to share my comments and suggestions with you. Overall, I must say that I appreciate the work you have put into this research, which I found both interesting and important. Taking a more holistic approach to the issue of the environmental impact of the consumption of digital content. Looking through your material, it seems as if the study is carried out with rigor, but I have some concerns with the data that you are relying on (more on that later, under Introduction in Part II). I have some general concerns with how the research is framed and presented in the paper, as well as some comments regarding your assumptions. The review is structured as follows. In Part I, I will give you some general comments concerning framing and underlying assumptions. In Part II, I will focus on more specific issues in each section of the paper.

Part I:

First, I want to focus on the discrepancy I find between the focus on individual consumption of digital content on the one hand, and the structural problems/changes needed to make this consumption more environmentally sustainable. In the paper, you are first very much concerned with how individuals use ICTs to consume digital content. You try to figure out who the "average user" is and calculate how much of the environmental "budget" such a user spend on consuming digital content. So far so good, I think. These findings are very interesting and they show in a very clear and effective way the actual environmental effects of these practices/behaviours. However, when you have established that and start discussing the actual "opportunities for mitigating the impacts of digital content consumption", you realize that many of these problems are global and structural problems, i.e., the individual agency is quite limited. You state that, obviously, using smaller devices for e.g. streaming is better than using a PC, but you do not explicitly say anything about how to transition or persuade individuals to use less energy and resource intensive devices. So there's a discrepancy between framing the problem on the individual level, but the solutions/source of the problems as largely structural. If the solutions/source of the problems are basically global or structural, what use is it then to measure the sustainability-related effects on the (average) individual level? I keep repeating myself here, but I don't see the point of dividing the effects to average individuals instead of focusing on the global "technomass" and the effects that accumulates in total, if you want to focus on the structural problems related to digital content distribution and consumption. As you have spent much time on finding the "average user", I think one way to overcome this discrepancy is to focus more on how structural changes can affect individual behaviours or social practices in the latter parts of the paper.

Second, I want to focus on the issue of unequal exchange of environmental effects. Because what you do when you try to find average users among the global population is that you obscure the fact that the effects that you present, concerning water and waste but also to some extent CO2 emissions, is the fact that people are not effected equally. ICTs are mainly used in the developed world for entertainment purposes but also to increase efficiency and accumulate wealth, but the negative effects of ICTs are oftentimes located in poorer countries and areas where ICTs are used to a much lesser extent (but this is where many of the precious materials used in ICTs are extracted). This is of course also related to the digital divide. A conclusion one might draw from your study is that Norwegian users do not need to change their consumption habits, while it is important that Indian users do (because of the electricity mix), however, doesn't the developed world have more responsibility, ethically speaking? Acknowledging this does not require you to rethink your study, however, I think it is an important issue to discuss.

Third, I would like to focus on some of your assumptions about the future. For example, you talk about the Agenda 2030, carbon neutrality by 2050 and the circular economy as targets/concepts that

will be achieved in the future. I can understand the 1,5C assumption (since otherwise you would not be able to come up with the individual carbon budget -- even though I am quite sure that we will miss that target completely), but assuming for example that circular economy will materialize is more controversial I think, especially when it comes to ICTs that consist of materials such as REEs that we don't know if we can recycle in the future. Even if this would be the case, assuming that a smartphone will last 100% longer in 2050 is difficult to say, since it is not the technology that is the limiting factor but the economic system that inherently require higher levels of consumption each year to function properly (this is why planned obsolescence is a thing). An interesting book on the issue called "Impossibilities of the Circular Economy" came out just a couple of years ago and problematizes such assumptions related to circularity. A discussion concerning your assumptions would thus be beneficial.

Part II:

Abstract

Clear and to the point. The last sentence could be rewritten or clarified. "behavioural change is paramount to prevent the increasing Internet demand from hindering sustainable lifestyles" -- does this imply consuming digital content in other ways (e.g., smaller devices) or does consumption of digital content promote other unsustainable lifestyle choices? It's not really clear what you are referring to here.

Introduction

"At present, an average individual spends daily seven hours on the Internet (>40% of the waking life)" -- Is this really true? I think you mean that the average INTERNET USER spends this much time online. Approximately 40 percent of the world's population is still not online at all. I think the source you are using only takes data from social media platforms, i.e., those who are not on these platforms are not in the sample (I assume). Please revise/check.

"a bottom up analysis considering the user's consumption patterns is still missing --in contrast to basic human needs like food and transport, already shown each to be responsible for ~20% of an individual's carbon footprint" I had to re-read this sentence a couple of times before I understood that the 20% referred to food/transport and not internet usage. Consider rewriting to clarify.

Results

Lines 58-63: You are using the same reference as for the "average individual", when you might in fact refer to the average individual who are actually USING internet. If this data is in fact describing what I think it does, you might need to remake some of your calculations. If you are assuming that these are the average numbers for an individual rather than for an internet user, your actual "worldwide consumption" might be approximately 40% higher than it should be! Please check this in your next version of the manuscript.

Lines 91-92: You have already said this before.

Lines 159-162: You say that video streaming has a big energy demand, which I assume is correct, but also that video streaming often takes place on energy demanding devices. I think you need to find a reference here. It would be nice to see how much of the video streaming that is being done on what device. My assumption is that streaming is more often done on larger screens (i.e., higher consumption) compared with social media usage, but that smartphones are still the go-to device also for video streaming. I might be wrong though, but it would be good to have a reliable source for your statement here.

Discussion and Methods

232-233: Why is metal extraction beyond the scope of the study? You are already looking into the CO2 emissions of energy production that is necessary to produce and power devices -- why would you not consider mining activities which is also on this high, structural level. Clarify why beyond scope.

235-239: What about the growing second-hand market (as a result of more expensive devices and components due to semiconductor shortages and inflation)? I think if you want to discuss prolonging the life of devices this is where you should put more focus in this discussion.

Response to Reviewers for NCOMMS-23-04500

We would like to express our gratitude to the Reviewers for the time spent in reviewing our manuscript and for their positive feedback. Their recognition of our work as a valuable contribution and their acknowledgement of the ICT's sector relevance in achieving environmental targets are deeply appreciated. We are particularly encouraged by the Reviewers' positive claims on the originality of our approach and noteworthy results.

We have taken the Reviewers' comments into careful consideration and have made appropriate changes to our manuscript to incorporate their suggestions effectively. In addition to addressing the specific comments of the Reviewers below, we have performed additional analyses to enhance the robustness of our results and conclusions. Specifically, we have now incorporated a sensitivity analysis to explore the impact of an increased renewable electricity procurement by data centres (Supplementary Figures 4 and 5) as well as additional prospective assessments considering alternative climate policy scenarios (Supplementary Figures 8, 9, and 10). Moreover, we have made substantial revisions to clarify the exploratory nature of our perspective analysis and the purpose it serves. We hope that these revisions satisfactorily address the specific concerns raised by the Reviewers and enhance the overall clarity and comprehensiveness of our manuscript.

The format adopted is as follows: comments in **blue**, replies in black, and actions in **bold**. Page and line numbers refer to the main article with highlighted changes.

Reviewer #1

The article is a valuable contribution, which presents the relevance of digital consumption for the achievement of environmental protection goals. What is particularly original is that the authors put the footprint in relation to the requirements for lifestyles that are in line with long-term environmental protection goals. Certainly, there are other areas that are more acute for environmental protection (e.g. food, transport, housing), and yet this paper shows impressively that the digital sector also has a contribution to make.

Still, I believe there is some room for improving the article further, which I list below.

Please note, that I can review climate-related impacts of digital technologies, and the results provided by the authors for this impact category seem plausible to me. However, I am not an expert in other impact categories (e.g. particulate matter, freshwater eutrophication) and cannot assess the plausibility of these results.

We greatly appreciate the Reviewer's insightful feedback on our article. We are pleased that our research is considered to be a valuable contribution. Moreover, we are happy to see that the Reviewer found our approach original, particularly concerning the contextualization of the environmental footprint of digital content consumption relative to sustainable lifestyles.

1. The abstract and the results section is missing reference years, e.g. "finding that web surfing, social media, video and music streaming, and video conferencing could consume on average ~40% of the per capita carbon budget consistent with limiting global warming to 1.5 °C,..."

We used 2020 as the reference year for the ICT technology-related data, whenever possible. For example, the electricity intensity of data centres was derived from information on the global data centres' electricity consumption and data traffic in 2020 based on refs. [1] and [2]. When 2020 data was unavailable, we extrapolated the most recent values to 2020 following sensible assumptions. For example, we could not find the electricity intensity of a smartphone for 2020, so we assumed an annual energy usage improvement of 3% to extrapolate the values from 2016 to 2020 based on ref. [3]. The same approach was applied to other end-user devices, as documented in Supplementary Table 5. Data from the ecoinvent database, which often consider several years, were assumed to be valid for the reference year 2020. On the other hand, digital content consumption patterns and user preferences refer to data collected for the period 2019-2022. This timeframe was chosen intentionally to prevent any distortion caused by the unusually high consumption levels during the COVID-19 pandemic. Overall, the reference year for technology data is here considered 2020, while consumption pattern may represent the current situation.

We have now explicitly mentioned that the findings correspond to the current situation (page 1, line 13; page 2, line 50; and page X, line X). To provide further clarity on the temporal scope, we have expanded the explanation of the data's year in the "Goal and scope definition" section in Methods, indicating that the reference year for technology data is 2020, while consumption behaviour and user's preferences are based on the latest available data for the period 2019-2022 (page 16, line 359).

[1] Masanet, E., Shehabi, A., Lei, N., Smith, S. & Koomey, J. Recalibrating global data center energy-use estimates. *Science* 367, 984–986 (2020)

[2] Cisco. *Cisco Global Cloud Index: Forecast and Methodology, 2016–202* (2018)

[3] Pärssinen, M., Kotila, M., Cuevas, R., Phansalkar, A. & Manner, J. Environmental impact assessment of online advertising. *Environmental Impact Assessment Review* 73, 177–200 (2018)

2. Also, you could make it more clear that your study takes a global perspective. E.g. in row 60 you talk about the "studied user". Is it the global average across all Internet users?

Yes, in this work we defined a "user archetype" based on the global average consumption patterns across all Internet users. Consumption patterns were defined based on the most recent surveys and statistics covering actual Internet users. **We have revisited the text to avoid ambiguity and clearly state that we defined a user archetype representing the global average consumption patterns across all Internet users (page 3, line 64).**

3. I would briefly mention in row 41+ that you conducted a LCA, and which standards you applied.

Thanks for this suggestion. **We have now clearly stated in the Introduction section that we applied the standardised life cycle assessment (LCA) methodology (page 2, line 45).**

4. In the abstract, you mention that behavioral change is required. This is very broad and should be specified.

We thank the Reviewer for highlighting this aspect. Accordingly, **we have revisited the abstract to explicitly indicate that we refer to a switch to less energy-intensive devices and settings (page 1, line 19)**, the environmental consequences of which are studied in our work.

5. You mention that consumers spend daily seven hours on the Internet. I am wondering if this is only active time spent on the Internet or also passive time spent on the Internet, e.g. streaming music in the background. I suppose, many devices are exchanging data with the "Internet" in the background, even though users conduct other activities. This could be specified.

We completely agree with the Reviewer that devices with Internet connection may be continuously exchanging data. More precisely, the figure of seven hours per day reported in our study refers to the daily time an average user spends consuming digital services. These include, for example, watching TV, using social media, and listening to music. We assume that this represents active time. Further details on this topic can be found in the original reference (Digital 2022 report: (<https://datareportal.com/reports/digital-2022-global-overview-report>)). **We have clarified this in the first paragraph of the manuscript, namely that "the average Internet user spends approximately seven hours per day active on the Internet..." (page 1, line 23).**

6. You state that only "recently" the environmental footprint of digital ICT gained interest. This might be true for the "public debate". However, ICT companies (such as telecommunication companies) and academia deal with this issue since at least 20 years. For example, see the studies by the Global e-Sustainability Initiative, Sustainability Reports by telcos such as Deutsche Telekom, or work published in the scope of the ICT for Sustainability or Environmental Informatics conference series.

Thanks for this clarification. **Following the Reviewer's suggestion, we have revised the text to mention explicitly that we refer to the "public interest" rather than the general interest (page 1, line 26).**

7. You state that the ICT sector causes roughly 1.2% of global anthropogenic GHG emissions. Recent meta-studies provide a different range, e.g.

<https://www.sciencedirect.com/science/article/pii/S2666389921001884>

<https://www.sciencedirect.com/science/article/pii/S0195925522002992>

We appreciate the Reviewer's suggestion regarding recent meta-studies of GHG emissions of the ICT sector. **We have updated the Introduction section to include the new range provided by the extensive review of Freitag et al., i.e., 1.0–1.7 Gt CO₂-eq or 1.2–2.8% of global anthropogenic GHG emissions (page 2, line 34).**

8. In the introduction, you mention the results of your study already. From my perspective, these should only be mentioned in the abstract or results section, see row 45+

Thanks for this comment. We acknowledge that it is common practice to present the results in the results section and the abstract. However, in accordance with the formatting instructions provided by *Nature Communications*, the final paragraph of the Introduction should contain a brief summary of the major results and conclusions. We could nevertheless modify this style if required.

9. "Norway exhibits the lowest impacts due to its highly renewable electricity mix, while the highest impacts are found in Australia (freshwater eutrophication), South Africa (marine eutrophication), China (particulate matter), and India (fossil resources use).  To me, it is unclear how the electricity mix relates to freshwater eutrophication, marine eutrophication, PM,....

The electricity mix affects the overall environmental footprint, including climate change and also other impact categories. In essence, following the standard LCA methodology, we consider the whole range of emissions and natural resources consumption throughout the life cycle of every power technology, i.e., the life cycle inventory (LCI) entries linked to 1 unit of power generated. This information, i.e., LCI entries of power generation, is further translated into 16 impact categories using suitable characterization factors. For example, Australia and Poland exhibit the highest impact in freshwater eutrophication due to their high reliance on lignite power plants. Notably, treatment of lignite mining spoil leads to emissions of phosphorus to groundwater, resulting in eutrophication. Moreover, the impact in marine eutrophication is slightly higher in countries with high shares of coal-based electricity, such as South Africa and India, due to higher emissions of nitrogen oxides and nitrates linked to coal combustion. **We have revised the text to clarify further the influence of the electricity mix on impacts beyond climate change (page 5, line 113).**

10. In Figure 3 you differentiate between operation and embodied impacts for all elements but the access and core network. To me, it is unclear why you do so and what life cycle phases have been considered in the access and core network. This also applies to Figure 4.

We have considered only the operational impacts associated with the access and core networks due to lack of data on infrastructure requirements (e.g., type and amount of equipment, i.e., routers and optical fiber, allocated per GB of data transferred). Yet, embodied impacts of the access and core network are deemed negligible compared with the use stage, as shown in previous studies [4,5]. To clarify this point further, **we have modified the legend in Figures 3 and 4 to explicitly refer to the operational impacts of the access and core networks.** Moreover, **in the Methods section we state that the manufacture and end-of-life management of the equipment required for the access and core networks have been omitted due to lack of data and significantly higher environmental importance of the use stage (page 17, line 394).** We hope that this clarifies our approach and reasoning.

[4] Malmodin, J. & Lundén, D. *The Energy and Carbon Footprint of the Global ICT and E&M Sectors 2010–2015. Sustainability* 10, 3027 (2018).

[5] Tao, Y., Steckel, D., Klemeš, J. J. & You, F. *Trend towards virtual and hybrid conferences may be an effective climate change mitigation strategy. Nat Commun* 12, 7324 (2021).

11. In Figure 4a, the black arrow is unclear to me.

The black arrow in Figure 4 indicates that the yellow dashed line refers to values on the right y-axis. We understand that this might be a bit convoluted. Hence, **we have modified Figure 4. In the revised figure, we have eliminated the black arrow and included the yellow dashed line in the legend with the following label: “Share of per capita carrying capacity (right y-axis)”.**

12. Potentially, the readability of the results section could benefit from some subheadings.

We have revised the results section to include more subheadings, which we believe will better guide readers through the key findings of our study. Specifically, **we now present the results displayed in Figure 2 in three new subsections. These are “Climate change impacts of digital content consumption”, which presents the results for carbon footprint and climate change, “Other environmental impacts associated with digital content consumption”, which shows the results for other impact categories such as acidification, eutrophication, and toxicity, and “Contribution analysis of environmental impacts”, which discusses the breakdown of impacts.**

13. In line 167+ you describe how you modeled the future scenario in regard to climate impacts. In line 174 you suddenly switch to other impact categories (e.g. freshwater eutrophication,...) without explaining how you modeled future scenarios for these. I assume these are not only related to the future climate scenarios.

Thanks for the comment. In the first part of our prospective analysis, we assess the impacts of digital content consumption under a future policy scenario compatible with limiting global warming to 1.5 °C. This scenario considers a range of technological changes that could be deployed to meet such climate goal. Specifically, the selected scenario is based on projected changes in electricity generation mixes. These include, for example, higher shares of renewables and bioenergy with carbon capture and storage (BECCS) to achieve net-zero electricity by 2050, and improvements in power generation efficiency, and cement and steel production, among others. Such technological changes will affect the entire environmental footprint of power generation, not only its carbon intensity. For example, shutting down coal power plants could help mitigate eutrophication and particulate matter formation, among others, due to lower emissions of nitrogen oxides and particulate matter. To improve clarity further, **we have revised the text and clarified that the gradual decrease in freshwater eutrophication, marine eutrophication, particulate matter formation, and ecotoxicity over time is due to the decline in fossil fuels usage according to the climate policy scenario considered (page 9, line 201).**

14. There are some grammatical/language errors in the paragraph in line 209+

Thank you for spotting this. **The text has been reviewed and the typos have been amended.**

15. Did you consider that demand for digital services could not only rise because of population growth and increasing access to digital services, but also due to rebound effects. As digital services become cheaper or more performant, demand per capita might also increase (e.g. cheaper access to faster broadband speeds often lead to higher demand for data traffic). This has been addressed in this paper with regard to 5G networks: https://link.springer.com/chapter/10.1007/978-3-031-18311-9_13
It is probably difficult to model these effects for the future, but they could be at least discussed.

Thanks for this insightful comment. In this work, we considered only current consumption patterns (e.g., number of hours spent active online). We did not attempt to project these data into the future, as such projections would bring about additional uncertainties. We acknowledge that the rebound effect and other factors could play a significant role in future consumption patterns and environmental impacts. Moreover, we agree that future research could explore these factors in greater detail. Accordingly, **we have now incorporated the rebound effect in the Supplementary Section on “Methodological assumptions, limitations, and future work”. Specifically, we acknowledge that the per capita demand for digital content could further increase in the future due to rebound effects (i.e., cheaper or more improved Internet access might result in higher data traffic), which would result in larger environmental footprints.**

16. I am aware that you did an uncertainty analysis with a Monte Carlo Simulation. And you state that your results do not change significantly, even if you consider the uncertainty. I am quite surprised that this is the case, given all the

uncertainty regarding life cycle inventory data of digital devices and infrastructures. There is large uncertainty regarding production impacts of end-user devices (which you addressed) but also regarding the general “energy intensity of the Internet”. For example, the study by the French Think Tank the Shift Project, was criticized for pessimistic assumptions regarding energy intensity (

<https://www.iea.org/commentaries/the-carbon-footprint-of-streaming-video-fact-checking-the-headlines>). The references 10 and 11 in the you use in the supplementary material are already quite old, even though I am not certain if more recent studies exist.

Even if you considered all sources of uncertainty, I think you should mention your results for a worst case (most pessimistic assumptions) and best case (most optimistic assumptions), as well as limitations in the availability of up-to-date data (if this is true), in the main text.

Thank you for this comment. As indicated in the manuscript, we conducted a Monte Carlo simulation to assess the robustness of our conclusions under uncertainties in the life cycle inventory (LCI) data (Supplementary Section 1.6). Here we distinguish between data of the foreground (main activities over which there is a specific level of control) and background systems (data of the surrounding processes), which combined provide the whole LCI. Accordingly, we modelled the probability distributions of the electricity intensity of electronic devices, equipment requirements, and data traffic demand (foreground data) based on the literature (Supplementary Tables 5 to 8). Moreover, we modeled the background data (i.e., manufacturing processes, power generation, etc.) with the default probability distributions available in ecoinvent v3.8.

The uncertainty analysis reveals that the shares of the per capita Earth’s carrying capacity linked to digital content consumption for acidification, terrestrial eutrophication, photochemical ozone formation, ozone depletion, carcinogenic human toxicity, ionising radiation, and land and freshwater use remain negligible, even after considering uncertainties in the LCI data (probabilities below 5% of shares above 2%). Regarding the most critical impact categories (shares >10% in the nominal scenario), the largest dispersion corresponds to mineral and metals resources depletion, freshwater eutrophication, particulate matter formation, non-carcinogenic human toxicity, and climate change. In the latter, most likely due to the shape of the lognormal distributions used to model the background data, the deterministic impact values presented in the main manuscript lies close to the lower bound of the uncertainty interval obtained in the uncertainty analysis. Hence, the required share of the carrying capacities in these categories is likely to be even higher than expected. Focusing on climate change, the uncertainty range (with outliers removed according to the 1.5 interquartile rule) of the required share of the per capita carrying capacity is 37-69%, while the deterministic value is 41%. Based on the uncertainty analysis, we can conclude that uncertainties associated with LCI are unlikely to alter the conclusions of our study.

To enhance the discussion of the uncertainty results, we have now represented the Monte Carlo simulation results by means of boxplots showing the interquartile range as well as the minimum and maximum values, with outliers removed (Supplementary Fig. 11). In response to the Reviewer's comments, we have expanded on the outcomes of the uncertainty analysis in the main text. We now explicitly indicate that we accounted for uncertainties in the LCI data. These include the uncertain data in the foreground system, particularly electricity intensity of electronic devices and equipment requirements, as well as in the background system modelled using ecoinvent (e.g., manufacturing processes and power generation). Hence, we have expanded on the uncertainty analysis results in the main text (page 14, line 317). Additionally, we have discussed the availability of up-to-date data, its implications on the results, and how we handle uncertainties through Monte Carlo simulation (Supplementary Section 4 on "Methodological assumptions, limitations, and future work").

17. You use different terms without explicitly stating what is meant by it, in particular the ICT sector, the Internet. If I understand correctly, you included end user devices, data centers and networks in your calculation. The aggregation of these components is usually called the ICT sector. However, not all data centers or network (usage) are part of the Internet, because the Internet is “just” the part of the infrastructure which is used for certain protocols (e.g. WWW). For example, some infrastructure (e.g. corporate data centers and networks) are used for other applications beyond

“the Internet”. I recommend clearly stating the system boundary and avoiding confusion between the two terms early in the article.

Thanks for this comment. We have revisited the manuscript to address this point. First, when defining the goal of the analysis, **we clearly state that we focus on quantifying the environmental impacts of digital content consumption encompassing all the necessary infrastructure linked to the consumption patterns of an average user (page 2, line 44)**. In the Results section, **we clearly define that we link the user consumption patterns to the natural resources required and emissions generated throughout the life cycle of the Internet network components, from raw materials extraction to manufacturing, distribution, operation, and end-of-life (page 3, line 75)**. Moreover, **we explicitly define that the system boundaries consider data centres, data transmission networks, customer premise equipment, and end-user devices (page 3, line 77)**. Finally, **we now systematically use the term “Internet network components” to refer to data centres, transmission network, and end-user devices**.

18. In my understanding, you used the regional average electricity mixes. Some ICT companies are more advanced in their use of electricity from renewable energies than the regional average. You could mention that in line 322.

Thank you for rising this insightful point. In our study, we indeed considered the electricity powering data centres to be supplied by a regional average electricity mix, weighted according to the share of data centres in each region. This implies that the global data centres are supplied by the regional electricity mixes in North America (39%), Asia (34%), Europe (22%), Latin America and the Caribbean (3%), and Middle East and Africa (2%).

We acknowledge that ICT companies are increasingly procuring renewable electricity, which can lead to reductions in the operational impacts of data centres. However, in our study, we opted to use regional average electricity mixes due to the challenges associated with accurately determining the proportion of global data centres powered by renewable electricity.

Nevertheless, we recognize the potential implications of this assumption and **have conducted a sensitivity analysis to address this concern. We considered six scenarios in which we varied the share of global data centres powered by renewable electricity ranging from 0% (current default assumption) to 100% (global data centres entirely powered by renewable electricity)**. The results, presented in **Supplementary Fig. 4**, reveals that **increased procurement of renewable electricity by data centres has the potential to mitigate certain impacts, particularly climate change, although the overall implications on our conclusions remain marginal**. For example, when considering that 100% of the global data centres are powered by wind, the global average share of the per capita carbon budget required by digital content consumption would decrease from 41% (using the regional electricity mix) to 31%. This relatively small reduction can be explained by the relatively smaller contribution of data centres to the overall life cycle impacts of digital content consumption, which are primarily dominated by the operational and embodied impacts of end-user devices (as shown in Fig. 3 in the main manuscript). Moreover, it is important to note that a 100% procurement of renewable electricity does not necessarily mean that data centres are supplied exclusively by renewable electricity, largely due to the variability of renewable energies. Hence, these results likely provide an upper bound on the mitigation potential of renewable electricity procurement. **We have now incorporated the results of the sensitivity analysis in the main text (page 17, line 384) and incorporated a discussion on this topic in the Supplementary Section 4 on "Methodological assumptions, limitations, and future work"**.

Reviewer #2

REVIEW OF “THE ROLE OF DIGITAL CONTENT CONSUMPTION IN ENVIRONMENTALLY SUSTAINABLE LIFESTYLES”

While this article tackles an important topic and it comes from a credible research team, it falls prey to several common errors and conveys illusory precision about the next few decades of information and communication technology (ICT) emissions. I recommend it be rejected in its current form.

It is a mistake to project emissions of computing technologies more than a few years ahead. Beyond typical manufacturer product roadmaps (which typically extend for 3-5 years), the characteristics of ICT technologies are impossible to predict. This article attempts to predict to 2050, which is absurd for ICT.

We appreciate the Reviewer's comment on the prospective analysis included in our work, and we value their perspective. However, while we recognize that it is hard to provide accurate future estimates, we fundamentally disagree on such a strong statement on the lack of value of prospective analyses. Assessing the future is always challenging due to the many uncertainties involved, but such prospective analyses are common practice as they provide valuable insights into likely trends and their broad implications. We stress that the goal here is not to predict the future accurately, but rather to shed light on different possibilities of future developments [1]. The literature is replete with studies that explore potential future environmental impacts, within the ICT sector (e.g., CO₂-saving potential of the metaverse up to 2050 [2] and video streaming regulation up to 2030 [3] and also beyond (e.g., transportation up to 2050 [4] and direct air capture technologies up to 2100 [5]). Underpinned by rigorous sensitivity analyses, we argue that such studies play a crucial role in guiding scientists and policymakers, facilitating future research, and shaping effective regulations.

Having said that, we understand the concerns raised by the Reviewer and **have made substantial revisions to further clarify the exploratory nature of our perspective analysis and the purpose it serves, and to explain the underlying methodology and assumptions. Additionally, we have performed additional sensitivity analyses to assess the robustness of our conclusions, finding that they are indeed quite robust.**

We would like to clarify further how the prospective analysis was conducted in this study. In essence, we used scenario analysis to examine how likely evolutions of key parameters may affect the environmental impacts of digital content consumption. In the results in Fig. 3, we consider future electricity generation mixes as well as improvements in power plants efficiency, steel and cement production, and the transport sector based on climate scenario results from the Integrated Assessment Model (IAM) IMAGE. Specifically, to model these background activities (e.g., electricity generation mixes), we assumed a climate scenario compatible with limiting global warming to 1.5 °C, in which the global electricity mix would become carbon neutral by 2050. The IAM data used in our work is available through the open-source software premise (<https://premise.readthedocs.io/en/latest/>). Importantly, ICT-related parameters, such as the electricity intensity of electronic devices or manufacturing processes, were kept constant at their current values, assuming 2020 as the reference year (as explained in the goal and scope definition section). This allows us to observe possible changes in the impacts of digital content consumption solely attributed to alterations in the background system, particularly electricity generation. **We have revisited the text as well as the caption of Fig. 3 to ensure the purpose of our prospective analysis and the underlying methodology and assumptions are fully clarified (page 9, line 189). Further explanation of the prospective analysis methodology is now included in the Methods (page 18, line 402).**

To further address the uncertainties surrounding the 1.5 °C scenario assumption, **we have performed new analyses considering two additional climate policy scenarios available from the IAM IMAGE: a baseline scenario compatible with limiting global warming to 3.5 °C (i.e., no decarbonization goals) and a scenario compatible with limiting global warming to 2 °C. These additional results, presented in Supplementary Fig. 8, have been incorporated in the main text to strengthen the robustness of our conclusions (page 9, line 210).** Overall, with this analysis we show that an aggressive climate policy compatible with limiting global warming to 1.5 °C or 2 °C could substantially reduce the climate impacts of digital content consumption. Meanwhile, other impacts mostly linked to raw materials extraction

and processing (e.g., gold mining) rather than energy consumption would require further improvements in the ICT sector.

Hence, in the results in Fig. 4 we explore changes in the ICT-related parameters, such as energy efficiency improvements and extended devices lifetime, in conjunction with the aforementioned changes in electricity generation and heavy industries. We used historical trends for energy efficiency improvements, while considering three hypothetical scenarios for lifetime extension (25%, 75%, and 100% lifetime extension). We acknowledge the high uncertainty associated with projecting these parameters into the future and explicitly present the results as what-if scenarios rather than precise impact predictions.

A key finding from this analysis is the higher importance of lifetime extension in mitigating impacts beyond climate change (i.e., eutrophication, particulate matter formation, ecotoxicity, and mineral and metal resources depletion) compared to energy efficiency improvements under a 1.5 °C climate scenario. As these impacts are not primarily linked to energy consumption, the environmental benefits of energy efficiency gains diminish in a decarbonized economy aligned with the 1.5 °C target. We further show that if the decarbonization of the economy aligns with the 1.5 °C target and the energy efficiency of electronic devices improves constantly and their lifetime doubles (100% lifetime extension), a substantial reduction of impacts linked to digital content consumption could be achieved by 2050, except for mineral and resource depletion. **We have revisited the text to acknowledge that the realization of this scenario entails substantial challenges associated with maintaining an energy efficiency improvement rate similar to historical trends and doubling the lifetime of electronic devices by 2050 (page 11, line 243). Moreover, we clearly state that the results provide an upper bound on the mitigation potential of these measures, highlighting that mineral and metal resources depletion is likely to remain a challenge (page 11, line 245).**

To further evaluate the robustness of these conclusions, **we repeated the analysis considering also the two additional climate policy scenarios (i.e., the baseline scenario compatible with limiting global warming to 3.5 °C and the scenario compatible with limiting global warming to 2 °C). These additional results, presented in Supplementary Fig. 9 and 10, confirm the large mitigation potential of lifetime extension.**

Lastly, we would like to stress again that our goal is not to make accurate predictions of how the future will look like, but rather to carry out an exploratory analysis on the broad environmental implications of the future ICT system considering plausible conditions. We believe that the qualitative conclusions derived from this exploratory analysis are robust, even considering the uncertainties affecting the calculations. Moreover, we believe that our revisions and additional analyses, intended to effectively address the Reviewer's concerns, have significantly enhanced the clarity and robustness of our study.

[1] National Research Council, 2010. *Persistent Forecasting of Disruptive Technologies*. National Academies Press, Washington, D.C. <https://doi.org/10.17226/12557>

[2] Zhao, N. & You, F. *The growing metaverse sector can reduce greenhouse gas emissions by 10 Gt CO₂e in the united states by 2050*. *Energy Environ. Sci.* (2023) doi:10.1039/D3EE00081H.

[3] Madlener, R., Sheykhha, S. & Briglauer, W. *The electricity- and CO₂-saving potentials offered by regulation of European video-streaming services*. *Energy Policy* 161, 112716 (2022).

[4] Zhang, C., Zhao, X., Sacchi, R. & You, F. *Trade-off between critical metal requirement and transportation decarbonization in automotive electrification*. *Nat Commun* 14, 1616 (2023).

[5] Qiu, Y. et al. *Environmental trade-offs of direct air capture technologies in climate change mitigation toward 2100*. *Nat Commun* 13, 3635 (2022).

This article also cites high-end estimates of ICT electricity use that are not credible, without explaining that context. For example, the work of Anders Andrae consistently overstates projected ICT electricity use, and when those growth rates don't materialize, Andrae just starts at the latest estimates of current electricity use (which is 3-5% for all ICT) and then projects rapid growth again, which always fails to materialize.

We acknowledge the concerns raised by the Reviewer about the credibility of some of the estimates of ICT electricity use. It should be noted that we did not use Andrae's work as a source of data. We cited such work as an example of studies that make "claims concerning the vast amounts of energy consumed by the information and communication technology (ICT) sector". Other studies, such as those by Jones (2018) and Belkir and Elmeligi (2018), have made similar claims. In our case, the electricity intensity of electronic devices was taken from various bottom-up studies, and its level of uncertainty was modelled using a probability distribution, as documented in Supplementary Table 5.

We have revisited the text to clarify that the claim about the vast amounts of energy consumed by the ICT sector comes from previous studies, rather than stating it as an established fact. More specifically, we specify that "Despite the ubiquity of the Internet in our lives, its environmental footprint started to gain public interest only recently, partially fuelled by studies claiming the substantial energy consumption of the information and communication technology (ICT) sector" (page 1 line 26).

There is a tendency to overestimate electricity use and emissions of ICT that is well documented [1]. One reason for that tendency is that forecasters ignore or underestimate the potential for efficiency improvements to offset most or all of growth in ICT service demand. As an example, data center services from 2010 to 2018 grew very rapidly (with compute instances, data flows, and data storage capacity growing manyfold), but total electricity use for data centers only grew 6% over that period [2]. Only in special cases like 2000 to 2005, when data center infrastructure was being built out for the first time in the US and Europe, does the data show very rapid growth in total electricity use by data centers [3].

Thank you for raising such an important aspect. In the analysis of the current impacts of digital content consumption displayed in Figures 2 and 3, we have taken 2020 as the reference year for ICT-related parameters, including the electricity intensity of servers, storage, and end-user devices. For data centres, we based our analysis on the electricity use of global data centres in 2020 as assessed in the reference 2 cited by the Reviewer. For other devices, if 2020 data was unavailable, we extrapolated the most recent data to 2020 by applying an annual energy usage improvement rate, as documented in Supplementary Table 5. For example, we assumed an annual energy usage improvement of 3% for smartphones, based on ref. [5]. While we acknowledge the uncertainties associated with extrapolation of historical trends, we believe that our approach is superior to simply ignoring energy efficiency gains, as highlighted in the reference 1 cited by the Reviewer.

In addition to explicitly considering efficiency improvements to extrapolate electricity usage to 2020, we explicitly modelled the uncertainties associated with life cycle inventory data (including electricity use and emissions) via the Monte Carlo sampling method. As mentioned above, the Monte Carlo simulation results reveal that uncertainties are substantial for critical impacts such as climate change, freshwater eutrophication, particulate matter formation, and mineral and metal resources depletion. Yet, the deterministic impact values (those generated for the nominal scenario) presented in the main manuscript appear to fall within the lower bound of the uncertainty range. This is most likely due to the shape of the lognormal distributions used to model the background data, as implemented in ecoinvent based on the Pedigree matrix. These results suggest that the required share of the carrying capacities is likely to be even higher. Consequently, we can conclude that uncertainties associated with LCI are unlikely to alter the qualitative outcome of our study. **In response to the Reviewer's comments, we have included additional text in the main manuscript to further elaborate on the main findings of the uncertainty analysis (page 14, line 317).**

[5] Pärssinen, M., Kotila, M., Cuevas, R., Phansalkar, A. & Manner, J. *Environmental impact assessment of online advertising. Environmental Impact Assessment Review* 73, 177–200 (2018)

Estimates of life cycle emissions for manufacturing of products always lags a lot, and so it's difficult to take seriously estimates from even a few years ago (because they are based on data from a few years before that), and things change so fast in the technology industry.

We acknowledge the inherent challenges in estimating life cycle emissions for manufacturing processes, especially considering the rapid pace of change in the ICT sector. In this work, we have retrieved the life cycle inventory (LCI) data

for electronic devices manufacturing from the ecoinvent database v3.8, which is regarded as the most comprehensive and widely used LCI database. However, we understand that some of the manufacturing data in ecoinvent may not capture the latest developments. For instance, the inventory for the production of a smartphone included in the latest version of the database is based on a 2014 LCA report. While we recognize the limitations of not using always up-to-date data, we believe that our uncertainty analysis, as presented in Supplementary Fig. 10 and the Discussion section, adequately captures the uncertainties associated with inventory data and their propagation into the results. Notably, the ecoinvent database uses the Pedigree matrix approach to model the uncertain inventory data based on lognormal distributions built using five independent data features: "reliability", "completeness", "temporal correlation", "geographic correlation", and "further technological correlation". This means that our uncertainty analysis explicitly accounts for the temporal variability of the manufacturing data, explicitly modelled in the Pedigree matrix. Importantly, the results of the Monte Carlo simulation indicate that the uncertainties associated with the LCI data are unlikely to significantly alter the main conclusions of our study.

We have taken great care to ensure the appropriate coverage of this point in the revised text, **explicitly stating that we considered the uncertainties associated with the LCI data, including manufacturing processes, and that we found that these uncertainties might not change the main outcome and conclusions of our analysis (page 14, line 317). We have improved the explanation of the Monte Carlo simulation results in the Supplementary Section 1.6, clearly mentioning the Pedigree matrix approach taken in the ecoinvent database. Moreover, we now discuss the limitations of not being able to always rely on up-to-date data (Supplementary Section 4 "Methodological assumptions, limitations, and future work").**

Understanding energy, emissions and materials impacts from ICT is important, but projecting to 2050 makes no sense. I suggest a more modest goal of accurately tallying current ICT impacts (including emissions, energy, and materials) for as late a year as the data support. That would be a contribution to the literature, as long as the caveats about the inadequacies of underlying data (particularly for embedded emissions and material use) are sufficiently well explained. That is a major reframing of this work, but if the authors are willing to undertake it, I would be willing to review it again.

REFERENCES

1. Koomey, Jonathan, and Eric Masanet. 2021. "Does not compute: Avoiding pitfalls assessing the Internet's energy and carbon impacts." *Joule*. vol. 5, no. 7. June 24. pp. 1625-1628.
[<https://www.sciencedirect.com/science/article/abs/pii/S2542435121002117>]
2. Masanet, Eric, Arman Shehabi, Nuo Lei, Sarah Smith, and Jonathan Koomey. 2020. "Recalibrating global data center energy-use estimates." *Science*. vol. 367, no. 6481. Feb 28. pp. 984.
[<http://science.sciencemag.org/content/367/6481/984.abstract>]
3. Koomey, Jonathan. 2008. "Worldwide electricity used in data centers." *Environmental Research Letters*. vol. 3, no. 034008. September 23. [<http://stacks.iop.org/1748-9326/3/034008>]

Once again, we appreciate the Reviewer's feedback on highly relevant aspects of our study. We want to emphasise the twofold goal of our study: firstly, to assess the current environmental impacts associated with digital content consumption and, secondly, to explore opportunities for impacts mitigation. Figures 2 and 3 show the impacts for a user archetype considering 2020 as the reference year. The prospective results presented in Figures 4 and 5 should be interpreted as an exploratory analysis that investigates several *what-if* scenarios. We hope that our response to the first comment now clarifies this point. We believe that the modifications we made and the additional results presented in the "Opportunities for mitigating the impacts of digital content consumption" section provide better clarity on this aspect. We still see much value in these prospective results, and the resulting conclusions whose robustness was carefully assessed via sensitivity analyses. However, if there are still concerns about this analysis, we would be open to moving it to the Supplementary Information.

Reviewer #3

This paper investigates the anthropogenic impact of users' digital content consumption across various environmental factors including climate, water, eutrophication, mineral and material use, etc. The authors consider the end-to-end impact of computing devices from raw material procurement to hardware manufacturing to operational use. A number of devices are considering including consumer electronics (e.g., smartphones, tablets, laptops), networking systems, and data centers.

The results of the analysis demonstrate a few noteworthy results:

The environmental impact of computing platforms owes not only to operational use but also raw material procurement and hardware manufacturing. In fact, in many cases, embodied environmental impacts outweigh the operational environmental impacts.

While ICT emissions currently account for a small portion of climate impact, if left unchecked, ICT will account for a large portion of per capita carrying capacity, in order to limit the global warming to within 1.5 degrees celsius in the coming decades.

A collection of efficiency improvements, lifetime extension, recycling, and behavioral change is needed to limit the climate impact of ICT.

Overall, I found the paper to be a very interesting read. While I have seen articles in the past that mention the embodied carbon of ICT emissions [1, 2] from the computer systems community, I have not seen the climate impact being put into context of the per capita carbon capacity of individuals and the world. This is a powerful result that can help encourage future research and investigation around sustainable computing. I also found the analysis going beyond carbon to include acidification, freshwater, eutrophication, particulate matter formation, fossil resource use, and land and water use to be helpful to understand a more holistic environmental footprint of ICT.

In terms of the methodology, the methodology adopted by the authors appears sound. There are a few assumptions and parameters that could be further substantiated or extended to explore additional scenarios (see below for questions) but overall the methodology seems appropriate. The use of life cycle analysis follows from established practices in the community and the comparison to per capita climate carrying capacity is based on targets from standardized communities and organizations. The supplemental information provides sufficient information to reproduce the results provided in the paper.

We sincerely appreciate the Reviewer's positive feedback. It is gratifying to hear that our research is deemed as a very interesting read. We are delighted that our study provides a unique perspective by contextualizing the environmental footprint of digital content consumption in relation to the per capita share of the Earth's ecological budget, a new angle yet to be fully explored. Additionally, we are glad that the inclusion of multiple environmental indicators beyond climate change, such as acidification, eutrophication, particulate matter formation, human toxicity, mineral and metal resources depletion, and land and water use, has been appreciated as it helps drawing a more holistic and comprehensive picture of the environmental footprint of Internet consumption. Thank you also for your positive assessment of our methodology, confirming its soundness. We are pleased to hear that our work and findings have the potential to inspire future research and investigations in the field.

There are a couple of specific questions I have for the authors:

In the introduction, you mention "data centers and data transmission networks jointly account for 2-3% of global electricity consumption" while "ICT sector [...] emitted 700 Mt CO₂-eq 2020 (equivalent to 1.2% of global anthropogenic greenhouse emission)". Could you expand on the gap between 1.2% of GHG emissions from ICT in total and 2-3% of electricity consumed by data centers alone? Furthermore, does the 1.2% of GHG emissions also include emissions from hardware manufacturing?

Thank you for bringing up this point. In response to your question, **we have updated the Introduction section as suggested by Reviewer #1 to include the latest figures for the GHG emissions of the ICT sector based on Freitag et al., i.e., 1.0–1.7 Gt CO₂-eq or 1.2–2.8% of global anthropogenic GHG emissions (page 2, line 34).** As indicated in the original reference, these values consider both operational and embodied emissions (e.g., from hardware

manufacturing). To address this point more clearly, **we have revisited the text and clarified that the cited GHG emissions of the ICT sector encompass both operational and embodied emissions (page 2, line 35).**

Regarding the electricity consumed by global data centres and data transmission networks (i.e., 2-3% of global electricity consumption as said in the Introduction), we wish to clarify that this information was obtained from the International Energy Agency. We believe that the discrepancies between the two values stem from the fact that the two ranges quantify different contributions. Specifically, they refer to different indicators (GHG emissions vs. electricity consumption), have distinct system boundaries (entire ICT sector vs. data centres and transmission networks), and are based on different data sources. **We have revisited the text to provide the two figures separately and to ensure that readers have a clear understanding of the different aspects being addressed (page 2, line 29).**

On page 4 you take the global average electricity mix (0.68 kg CO₂-eq per kWh). However, as you mention in your methodology, this does not take into account the distribution of users and use patterns across the world (for instance, you say the remaining 40% of unconnected users in the world come largely from India and China which have higher carbon intensity than the world average). How sensitive are your results to the electricity mix and use patterns across the world?

Thank you for this comment. We would like to clarify that we have calculated the environmental impacts of digital content consumption considering both the global average electricity mix as well as the national electricity mix for a number of countries, as shown in Figure 2. As noted in the main manuscript, *“the carbon footprint of digital content consumption varies significantly with the user’s location, ranging from 146 kg CO₂-eq year⁻¹ in Norway to 327 kg CO₂-eq year⁻¹ in India”* (page 4, line 86). Moreover, we indicate that *“Such variability is due to the GHG emission intensity of the electricity mix in each country, with Norway relying mostly on hydropower (0.02 kg CO₂-eq kWh⁻¹) and India on fossil fuels (1.34 kg CO₂-eq kWh⁻¹)”* (page 4, line 87).

Unfortunately, we did not analyse the variability in use patterns across countries due to lack of data. Instead, we assumed the same use patterns for each location, which is based on the global average across all Internet users. However, it is worth noting that we explored two alternative user archetypes that differ in their preferences (e.g., the type of devices used). **The revised section on “Opportunities for mitigating the impacts of digital content consumption” present the results for the two alternative user archetypes (page 8, line 177). For example, the results show that switching to less demanding end-user devices and settings can reduce the most critical impacts by 53-67%. Moreover, the implications of the lack of data on users’ consumption patterns across different countries is now highlighted in the Supplementary Section 4 on “Methodological assumptions, limitations, and future work”.**

On page 8 you mention *“the embodied impacts of the IT equipment used in data centers (e.g., servers and storage equipment) are less relevant”*. Why is this? In fact, for many hyperscale data centers (e.g., Google, Meta, Microsoft, Amazon) that are procuring an increasing amount of renewable energy to power their operation, the majority of their environmental impact owes to embodied emissions [1, 2]. Can this impact be taken into account? Or, should the reader see the estimates as a lower bound for ICT’s impact.

We considered both the operational impacts of data centres as well as the embodied impacts due to the extraction of raw materials, manufacturing, distribution, and end-of-life disposal of servers and storage equipment. In our study, the embodied impacts of data centres were found to be less relevant compared with the operational impacts (<2% of the total impact of digital content consumption across all the categories). The higher relevance of the operational stage is confirmed by prior LCA studies on data centres, as reviewed by the UNEP (2020) [1]. In our study, these results can be explained by two main reasons. Firstly, we considered the regional distribution of global data centres. Accordingly, the electricity required by data centres is supplied by the regional mix in North America (39%), Asia (34%), Europe (22%), Latin America and the Caribbean (3%), and Middle East and Africa (2%). These mixes currently rely heavily on fossil fuels, making the operation of data centres the main driver of impacts such as climate change. Secondly, we modelled the embodied impacts of IT equipment per unit of data traffic considering the lifetime of the equipment (i.e., 5.5 years on average). In other words, the embodied impacts are distributed over the lifetime of that equipment, resulting in

relatively small impacts per unit of data traffic. The distribution of embodied impacts over the infrastructure's lifetime is common practice in LCA (refer to [2] for an example for data centres).

However, we agree with the Reviewer that some data centres may cover part of their electricity demand with on-site renewable energy sources, with potential implications on their environmental footprint. In response also to Reviewer #1, **we have conducted a sensitivity analysis (refer to Supplementary Fig. 4). We have found that increased procurement of renewable electricity by data centres has the potential to mitigate certain impacts, particularly climate change, although the overall implications on our conclusions remain marginal.** For example, when considering that 100% of the global data centres are powered by wind, the global average share of the per capita carbon budget required by digital content consumption would decrease from 41% (using the regional electricity mix) to 31%. This relatively small reduction can be explained by the relatively smaller contribution of data centres to the overall life cycle impacts of digital content consumption, which are primarily dominated by the operational and embodied impacts of end-user devices (as shown in Fig. 3 in the main manuscript). **Moreover, we provide the breakdown of data centres' impacts in Supplementary Fig. 5, showing that embodied impacts could represent as much as 26% and 80% of the data centres' climate impacts and mineral and resources depletion impacts under the assumption of 100% wind-powered data centres.**

[1] UNEP: *Environmental sustainability of data centres: A need for a multi-impact and life cycle approach* (2020).

[2] Whitehead, B., Andrews, D. & Shah, A. *The life cycle assessment of a UK data centre. Int J Life Cycle Assess* 20, 332–349 (2015).

Following from the previous point Figure 3 considers operational and embodied emissions for all components except the access and core networks. Why are these not considered? Or is there insufficient data to estimate the embodied impacts of the networking systems?

We have considered only the operational impacts associated with the access and core networks due to lack of data on infrastructure requirements (e.g., how much equipment –routers and optical fibre– should be allocated per GB of data transferred). Yet, embodied impacts of the access and core network are deemed negligible compared with the use stage, as shown in previous studies [3,4]. To clarify this point, **we have modified the legend in Figures 3 and 4 to explicitly refer to the operational impacts of the access and core networks.** Moreover, **in the Methods section we state that “The equipment required for the access and core networks have been omitted due to the lack of data and the significantly higher environmental importance of the use stage” (page 17, line 394).** We hope that this clarifies our approach and reasoning.

[3] Malmodin, J. & Lundén, D. *The Energy and Carbon Footprint of the Global ICT and E&M Sectors 2010–2015. Sustainability* 10, 3027 (2018).

[4] Tao, Y., Steckel, D., Klemeš, J. J. & You, F. *Trend towards virtual and hybrid conferences may be an effective climate change mitigation strategy. Nat Commun* 12, 7324 (2021).

It is unclear in Figure 4 if “climate impact” (top left plot) is solely from digital content consumption (the operational emissions from running devices) or also from manufacturing the devices used. The analysis also assumes a high degree of carbon capture technology which has not been demonstrated to be effective at a large scale. Would it be possible to have two scenarios, with an optimistic and more pessimistic view of renewable energy/carbon capture technology? Without this context, I wonder if the conclusion is “efficiency” and carbon-optimization for ICT is not needed given renewable energy and carbon capture will eliminate the climate impact of ICT devices.

The impacts displayed in Figure 4 include both operational and embodied emissions for end-user devices, customer premise equipment, access and core network (only operational), and data centres. The different colours correspond to each one of these categories, as also indicated in the legend of the figure.

The Reviewer raises an interesting point regarding the climate policy scenario considered, which is compatible with limiting global warming to 1.5 °C and, consequently, assumes a high degree of renewable energies and carbon capture and storage. We agree with the Reviewer that having additional climate scenarios would increase the robustness of our

conclusions. In this regard, **we generated new results for the impacts of digital content consumption under two additional climate policy scenarios: one scenario compatible with limiting global warming to 3.5 °C (baseline scenario, no decarbonization goals) and another one compatible with the less ambitious 2 °C target.** The results, presented in Supplementary Figures 8, 9, and 10, show that a climate scenario compatible with the 2 °C target would achieve similar impact reductions than the 1.5 °C scenario. In contrast, the baseline scenario entails a low mitigation of the climate impacts associated with digital content consumption by 2050, while exacerbating other impacts such as freshwater eutrophication and ecotoxicity (relative to the 1.5 °C or 2 °C scenarios). This further implies that energy efficiency gains would have a higher mitigation potential in the baseline scenario than under the 1.5 °C or 2 °C scenarios. **We have included additional text in the main manuscript to further elaborate on the main findings of these assessments (page 9 line 210).**

Once again, overall I feel the paper makes strong contributions to understand the holistic environmental impact of technology and motivates future work. I hope the authors can address the questions above to clarify some of the details around the work!

[1] Chasing Carbon: The Elusive Environmental Footprint of Computing

[2] The Dirty Secret of SSD's: Embodied Carbon

[3] GreenChip: A tool for evaluating holistic sustainability of modern computing systems

We are delighted to hear these positive comments on our potential contribution to understanding the holistic environmental impacts of the Internet, which could motivate future research in this field. We highly value the Reviewer's comments, as they provide valuable insights and opportunities for further clarification and improvement. We have carefully addressed each of the comments raised and ensured that the details around our work are clarified in the revised version of the manuscript. Thank you once again for your thoughtful review.

Reviewer #4

Dear author(s) and/or editor(s),

I would like to thank you for the opportunity to review your manuscript "The role of digital content consumption in environmentally sustainable lifestyles". I have now completed my review and would like to share my comments and suggestions with you. Overall, I must say that I appreciate the work you have put into this research, which I found both interesting and important. Taking a more holistic approach to the issue of the environmental impact of the consumption of digital content. Looking through your material, it seems as if the study is carried out with rigor, but I have some concerns with the data that you are relying on (more on that later, under Introduction in Part II). I have some general concerns with how the research is framed and presented in the paper, as well as some comments regarding your assumptions. The review is structured as follows. In Part I, I will give you some general comments concerning framing and underlying assumptions. In Part II, I will focus on more specific issues in each section of the paper.

We would like to express our gratitude to the Reviewer for taking the time to review our manuscript and for the valuable feedback. We are glad to hear that our research has been found both interesting and important, reflecting the effort and dedication we have invested in this study. We appreciate the recognition of our holistic approach to examining the environmental footprint of digital content consumption. Thank you once again for your thoughtful review and for highlighting the significance of our research.

Part I:

First, I want to focus on the discrepancy I find between the focus on individual consumption of digital content on the one hand, and the structural problems/changes needed to make this consumption more environmentally sustainable. In the paper, you are first very much concerned with how individuals use ICTs to consume digital content. You try to figure out who the "average user" is and calculate how much of the environmental "budget" such a user spend on consuming digital content. So far so good, I think. These findings are very interesting and they show in a very clear and effective way the actual environmental effects of these practices/behaviours. However, when you have established that and start discussing the actual "opportunities for mitigating the impacts of digital content consumption", you realize that many of these problems are global and structural problems, i.e., the individual agency is quite limited. You state that, obviously, using smaller devices for e.g. streaming is better than using a PC, but you do not explicitly say anything about how to transition or persuade individuals to use less energy and resource intensive devices. So there's a discrepancy between framing the problem on the individual level, but the solutions/source of the problems as largely structural. If the solutions/source of the problems are basically global or structural, what use is it then to measure the sustainability-related effects on the (average) individual level? I keep repeating myself here, but I don't see the point of dividing the effects to average individuals instead of focusing on the global "technomass" and the effects that accumulates in total, if you want to focus on the structural problems related to digital content distribution and consumption. As you have spent much time on finding the "average user", I think one way to overcome this discrepancy is to focus more on how structural changes can affect individual behaviours or social practices in the latter parts of the paper.

The Reviewer raises a very interesting point with deep and fundamental implications. We strongly believe that the path to sustainable development would require both, bottom-up and top-down efforts. Bottom-up approaches that focus on individuals' consumption patterns and per capita ecological limits could effectively guide decisions towards sustainable lifestyles. These efforts should be complemented by top-down actions, including regulations and further technological developments. Our proposed framework assesses the environmental consequences of both behavioural (e.g., in terms of preferences regarding devices, etc.) and technological changes (e.g., decarbonization of power mixes, efficiency improvements in devices, etc.). Specifically, we investigated behavioural changes by exploring two alternative user archetypes that differ in their preferences (refer to Supplementary Figs. 6 and 7). However, we acknowledge that these results and their implications were not discussed in the most effective manner. We agree with the Reviewer's observation that the role of user's preferences could get diluted in the discussion about energy system decarbonization and other technological changes.

To avoid this, the text has been revisited to implement the following modifications. **Firstly, the revised section on “Opportunities for mitigating the impacts of digital content consumption” now addresses how both behavioural and technological changes may contribute to covering our Internet demand more sustainably (page 8, line 177). In this regard, a paragraph presenting the results for the two alternative user archetypes is now included at the beginning of the section (page 8, line 177).** Here, we show that a switch to less demanding end-user devices and settings can reduce the most critical impacts by 53-67%, with large implications for the required share of the per capita Earth’s carrying capacity. However, we also show that a switch to the most demanding devices (i.e., 4K video streaming, high quality video meetings, etc.) would increase the most critical impacts by 21-31%. Hence, with these results we demonstrate the potentially large influence of the user’s preference on the environmental impacts.

Furthermore, we now included a paragraph in the Discussion section highlighting the role of behavioural changes to mitigate environmental impacts along with a reflection on effective ways to implement such changes (page 14, line 296). More specifically, we now highlight that behavioural change can potentially mitigate impacts earlier and at lower cost than technological changes; however, how to devise effective ways to encourage consumers to shift to more sustainable behaviours remains challenging. We then elaborate further on some possible ways to accomplish this goal, namely to raise awareness among Internet users regarding (i) the associated environmental impacts, and (ii) the potential benefits of regulatory interventions targeting the users’ behaviour. We hope that the implemented modifications have addressed the Reviewer’s concern.

Second, I want to focus on the issue of unequal exchange of environmental effects. Because what you do when you try to find average users among the global population is that you obscure the fact that the effects that you present, concerning water and waste but also to some extent CO2 emissions, is the fact that people are not effected equally. ICTs are mainly used in the developed world for entertainment purposes but also to increase efficiency and accumulate wealth, but the negative effects of ICTs are oftentimes located in poorer countries and areas where ICTs are used to a much lesser extent (but this is where many of the precious materials used in ICTs are extracted). This is of course also related to the digital divide. A conclusion one might draw from your study is that Norwegian users do not need to change their consumption habits, while it is important that Indian users do (because of the electricity mix), however, doesn't the developed world have more responsibility, ethically speaking? Acknowledging this does not require you to rethink your study, however, I think it is an important issue to discuss.

Thanks for this insightful and highly relevant comment. The Reviewer is right in pointing out that impacts will affect regions in various ways. Even if the impact is global, like climate change, the consequences will be asymmetric across the globe. For example, sub-Saharan Africa and Asia are expected to suffer the most severe health effects associated with climate change (primarily undernutrition and infectious disease) [1]. Likewise, some impacts have a strong regional component, like those connected to nutrients, toxic, and particulate matter emissions linked to the mining and processing of metals for electronic devices. Focusing on climate change, the same fundamental ethical question arises when deciding on which countries should primarily carry the burden of mitigating emissions (or even deploying negative emissions technologies [2]). Here we assumed an egalitarian sharing principle of the Earth’s carrying capacity, acknowledging that other principles (e.g., equality, needs, right to development, sovereignty, and capability) could have been applied instead [3].

In response to this valuable feedback, **we have expanded the Discussion section to delve more deeply into this issue (page 14, line 308).** We now explicitly state that, while it is important for all users to embrace more sustainable consumption habits and for the energy and ICT sectors to reduce emissions, ensuring sustainable development could be approached through various principles. We acknowledge that, in practice, developed countries are likely to play a leading role and accelerate their decarbonization efforts due to their historical emissions and to compensate for slower low-carbon transitions in developing countries [4]. Additionally, we recognize that the allocation principle to share the Earth’s carrying capacity is a much-debated topic, and alternative allocation principles have been proposed in the literature without reaching yet any consensus on them.

[1] World Health Organization. *Quantitative risk assessment of the effects of climate change on selected causes of death, 2030s and 2050s* (2014).

[2] Cobo, S., Galán-Martín, Á., Tulus, V., Huijbregts, M. A. J. & Guillén-Gosálbez, G. *Human and planetary health implications of negative emissions technologies*. *Nat Commun* 13, 2535 (2022).

[3] European Environment Agency. *Is Europe living within the limits of our planet — An assessment of Europe's environmental footprints in relation to planetary boundaries* (2020).

[4] Rissman, J. et al. *Technologies and policies to decarbonize global industry: Review and assessment of mitigation drivers through 2070*. *Applied Energy* 266, 114848 (2020).

Third, I would like to focus on some of your assumptions about the future. For example, you talk about the Agenda 2030, carbon neutrality by 2050 and the circular economy as targets/concepts that will be achieved in the future. I can understand the 1.5C assumption (since otherwise you would not be able to come up with the individual carbon budget -- even though I am quite sure that we will miss that target completely), but assuming for example that circular economy will materialize is more controversial I think, especially when it comes to ICTs that consist of materials such as REEs that we don't know if we can recycle in the future. Even if this would be the case, assuming that a smartphone will last 100% longer in 2050 is difficult to say, since it is not the technology that is the limiting factor but the economic system that inherently require higher levels of consumption each year to function properly (this is why planned obsolescence is a thing). An interesting book on the issue called "Impossibilities of the Circular Economy" came out just a couple of years ago and problematizes such assumptions related to circularity. A discussion concerning your assumptions would thus be beneficial.

We appreciate the valuable comment from the Reviewer, and we recognize the inherent uncertainties in modelling future scenarios. To address these concerns and enhance the robustness of the analysis, we have taken additional steps in our study as outlined below.

Firstly, in response to the uncertainties surrounding the 1.5 °C scenario assumption, **we have incorporated two additional climate policy scenarios: a baseline scenario compatible with limiting global warming to 3.5 °C (i.e., no decarbonization goals) and a scenario compatible with limiting global warming to 2 °C. The results for these scenarios, presented in Supplementary Figures 8, 9, and 10, show that the 2 °C scenario has similar impacts to the 1.5 °C scenario, whereas the baseline scenario shows negligible mitigation of the climate impacts while exacerbating other impacts such as freshwater eutrophication and ecotoxicity. We have now included these results in the main text to strengthen the robustness of our conclusions (page 9, line 210).**

Secondly, we fully acknowledge the challenges related to the materialization of the circular economy in the ICT sector and beyond. To account for these uncertainties, we considered three exploratory scenarios for lifetime extension, namely 25%, 75%, and 100% extension. These scenarios aim to shed light on future trends and their implications and are not intended to be taken as accurate impact predictions. Thus, we consider these results as what-if scenarios to provide insights into the mitigation potential of the said circularity measures. Therefore, **we have revisited the text to acknowledge that the realization of this scenario entails substantial challenges associated with doubling the lifetime of electronic devices by 2050. Moreover, we also state that the results provide an upper bound on the mitigation potential of these measures, highlighting that mineral and metal resources depletion could remain challenging (page 11, line 243).**

We appreciate the Reviewer's insights, and we believe that the inclusion of alternative climate policy scenarios and the acknowledgment of the challenges associated with circular economy assumptions strengthen the comprehensiveness and robustness of our study.

Part II:

Abstract

Clear and to the point. The last sentence could be rewritten or clarified. "behavioural change is paramount to prevent the increasing Internet demand from hindering sustainable lifestyles" -- does this imply consuming digital content in

other ways (e.g., smaller devices) or does consumption of digital content promote other unsustainable lifestyle choices? It's not really clear what you are referring to here.

Here, behavioural change is linked to our analysis of alternative user archetypes. Hence, it means using less energy-intensive end-user devices and settings (e.g., using only a smartphone and watching video streaming at low quality). **We have revisited the abstract to explicitly mention that we refer to a switch towards less energy-intensive devices and settings (page 1, line 19).**

Introduction

"At present, an average individual spends daily seven hours on the Internet (>40% of the waking life)" -- Is this really true? I think you mean that the average INTERNET USER spends this much time online. Approximately 40 percent of the world's population is still not online at all. I think the source you are using only takes data from social media platforms, i.e., those who are not on these platforms are not in the sample (I assume). Please revise/check.

Thank you for bringing this to our attention. The Reviewer is right, we should have specified the average Internet user, not the average individual. **We have revised the statement accordingly to avoid any confusion (page 1, line 23).** To clarify further, please note that the figure of seven hours per day reported in our manuscript refers to the daily time an average user spends consuming digital services. These include, for example, watching TV, and using social media, but also listening to music. We assume that all these activities represent active time. Further details are reported in the original reference (Digital 2022 report: (<https://datareportal.com/reports/digital-2022-global-overview-report>)).

"a bottom up analysis considering the user's consumption patterns is still missing –in contrast to basic human needs like food and transport, already shown each to be responsible for ~20% of an individual's carbon footprint" I had to re-read this sentence a couple of times before I understood that the 20% referred to food/transport and not internet usage. Consider rewriting to clarify.

The sentence has been rewritten in a more concise way (page 2, line 39). We meant originally that prior studies found that basic human needs like food and transport contribute 20% each to the per capita carbon footprint. Meanwhile, to our best knowledge, the contribution of Internet consumption to the per capita carbon footprint (and other impact categories) remains unexplored.

Results

Lines 58-63: You are using the same reference as for the "average individual", when you might in fact refer to the average individual who are actually USING internet. If this data is in fact describing what I think it does, you might need to remake some of your calculations. If you are assuming that these are the average numbers for an individual rather than for an internet user, your actual "worldwide consumption" might be approximately 40% higher than it should be! Please check this in your next version of the manuscript.

We thank the Reviewer for highlighting this important aspect. Here, we defined a "user archetype" based on the global average consumption patterns across all Internet users. Consumption patterns were defined based on the most recent surveys and statistics covering actual Internet users. For example, we estimated the annual consumption of social media (894 hours per year) from the average daily time spent with social media by Internet users aged 16 to 64 (Digital 2022 report: (<https://datareportal.com/reports/digital-2022-global-overview-report>)). We are assuming that these are the average numbers for an Internet user. Hence, our global average impacts correspond to the impacts generated by the global average Internet user. **We have revisited the text to avoid ambiguity and clearly state that we defined a user archetype representing the global average consumption patterns across all Internet users (page 3, line 64).**

Lines 91-92: You have already said this before.

The text has been changed to avoid repetitive information.

Lines 159-162: You say that video streaming has a big energy demand, which I assume is correct, but also that video streaming often takes place on energy demanding devices. I think you need to find a reference here. It would be nice to see how much of the video streaming that is being done on what device. My assumption is that streaming is more often

done on larger screens (i.e., higher consumption) compared with social media usage, but that smartphones are still the go-to device also for video streaming. I might be wrong though, but it would be good to have a reliable source for your statement here.

Thanks for raising this point, which was already included in the analysis via data from a survey of global online video viewers carried out in 2019 and published by Statista [1]. More precisely, we used recent statistics on users' preferences in the said reference to determine the device type used to access digital content. As shown in Supplementary Table 4, video streaming is shared among smartphones (ca. 31% of the annual consumption), TVs (25%), tablets (16%), laptops (14%), and desktop computers (14%). Moreover, video streaming resolution is equally split into 720p and 1080p. In order to clarify this point, **we have now specified that we considered that TVs, desktop computers, and laptops are used about 53% of the time, while less energy intensive smartphones and tablets account for 47% (page 8, line 173). Moreover, the reference to the Statista survey and Supplementary Table 4 has been included.**

[1] <https://www.statista.com/statistics/784351/online-video-devices/#:~:text=According%20to%20an%20August%202019,TVs%20and%20other%20connected%20devices.>

Discussion and Methods

232-233: Why is metal extraction beyond the scope of the study? You are already looking into the CO₂ emissions of energy production that is necessary to produce and power devices -- why would you not consider mining activities which is also on this high, structural level. Clarify why beyond scope.

We considered the impacts from metal extraction and processing based on current practices. However, what we meant by "beyond the scope of the study" is that we did not explicitly assess the potential reduction in emissions resulting from improvements in mining practices. While we acknowledge that improvements in mining practices could contribute to reducing the life cycle impacts of Internet consumption, we preferred to omit them due to data gaps and the additional uncertainties they would bring about. Nonetheless, we recognize the potential importance of this aspect and believe that it should be addressed in future research.

235-239: What about the growing second-hand market (as a result of more expensive devices and components due to semiconductor shortages and inflation)? I think if you want to discuss prolonging the life of devices this is where you should put more focus in this discussion.

Thanks for this suggestion. **We now included this aspect in the discussion section and linked it to the role of a more circular economy, namely that *"An increased reutilization of electronic devices, e.g., via the second-hand market, could contribute notably to lifetime extension and embodied impacts mitigation. Yet, the materialization of reuse strategies will ultimately depend on consumer preferences and price attractiveness over a new device"* (page 13, line 288).**

Reviewers' Comments:

Reviewer #1:

Remarks to the Author:

Thanks for the opportunity to review the article again. The authors have adequately addressed all my comments from the first round.

Overall, the article is now much clearer. The article still makes an original contribution by putting the footprint of digital consumption in relation to the earth's carrying capacity.

I am aware that the analysis and future scenarios are subject to many uncertainties. The authors have addressed the most critical uncertainties and the overall results seem plausible.

However, there is one point I want to raise: When you provide country-specific results in line 88+, you consider differences in electricity mixes across countries. I assume regional differences in actual digital content consumption would also impact the results but have not been accounted for, as you focus on the "global average user". I am aware that considering diverging consumption levels across countries would add a lot of additional complexity. You mention this issue in the discussion section. However, by using a global average, many countries in the Global South are unfairly credited with high digital consumption that doesn't exist in reality. If these countries now have a CO₂-intensive electricity mix, they perform very poorly in a country comparison. Is there any way to make a rough estimate (even just an example) to address this uncertainty?

Other than that, the article is ready for publication, in my opinion.

Reviewer #2:

Remarks to the Author:

SECOND REVIEW OF "THE ROLE OF DIGITAL CONTENT CONSUMPTION IN ENVIRONMENTALLY SUSTAINABLE LIFESTYLES" (REVIEWER #2)

There are two main issues raised by reviewers that the responses of the authors fail to capture. The first is that it is absurd to estimate energy and emissions for information technology (IT) to 2050. The second (raised by Reviewer 4 and with which I concur) is that this article frames the issues around emissions of IT as a problem of individual behavior rather than a problem that requires systemic response.

On the first key issue, the authors can do all the sensitivities they want, and state that they are just exploring alternative futures, but it's just not possible to say anything sensible about ICT electricity use and emissions two or three decades from now. Imagine someone in the year 2000 trying to predict what would happen in the next 23 years. Could they have predicted the build out of data center infrastructure from 2000 to 2005, the rise of the Internet of things, the dominance of handheld computing (phones), the emergence of hyperscale data centers, or the recent rush to machine learning? I don't think so. Things change too fast in IT, and I don't think it helps when researchers who apparently don't understand this issue try to create scenarios for several decades hence. Ultimately this is an issue for the editor to decide, but if it were up to me, I wouldn't publish research that embodies the flawed idea that projecting several decades hence makes any sense in this context.

On the second main issue, the title of the article tells the tale: "The role of digital content consumption in environmentally sustainable lifestyles". It's not about content consumption and it's not about lifestyles, it's about a system that people use to apply IT to their lives. To first order, people's choices of which content to consume DO NOT MATTER to the electricity use and emissions associated with the IT system, and it is a mistake to imply that this is fundamentally a problem of consumption and

lifestyles. For example, the energy use of routers, switches, and other network equipment doesn't change much when these devices are either running flat out or not being used at all. The energy use of most computers is dominated by standby and off power, which are also not affected much by consumption or lifestyle choices. So the framing of this article is all wrong.

I think the authors have done some useful work and could say something important about nearer term questions. If they just focused on 2030 instead of 2050 and moved away from implying that ICT emissions are mainly a subject for individual consumption and lifestyle choices, then I could get behind publishing it. But as it stands, I recommend that it not be published in its current form.

Reviewer #4:

Remarks to the Author:

I would first like to thank the authors for the very detailed responses to our feedback on your initial draft. It makes our work much easier and shows that you have put effort and consideration into this new version of the manuscript. Overall, I feel that your manuscript is improving, although I still have some general comments that I would like you to consider in your next round of revisions.

While I value the notion of assigning individuals an annual carbon budget for allocation across diverse ICT-related activities, along with the breakdown of carbon consumption linked to specific ICT applications like video streaming, I do acknowledge the presence of certain inherent constraints within this approach. These acknowledged limitations underscore the need for careful consideration. For example, as I also emphasized last time, it implies that a Norwegian user uses much less of their carbon budget to consume digital content than an Indian user. You clearly describe this in the paper, but I feel that there are problematic implications of this fact are not properly discussed. Either it implies that Norwegian (or Western in general) users should be "allowed" to consume much more digital content than for example Indian users (by a great deal, too). This means that reducing the ICT usage in developing countries would be beneficial for the climate, but in many developing countries (such as Norway or Sweden) the climate impact of these activities is almost negligible. This is the harsh reality as of right now, with the current electricity mix in these two contexts being radically different, but stands in stark contrast with your statement: "While it is crucial for all users to embrace more sustainable consumption habits and for the energy and ICT sectors to reduce emissions worldwide, the endeavour to ensure sustainable development can be approached through various principles. In practice, developed countries are likely to play a leading role and accelerate their decarbonisation efforts to compensate for the lower decarbonisation pace in developing countries". Alternatively, it implies that we should in fact use different budgets for different contexts, but I am not sure how such a budget would be designed. In practice, focusing on either lowering the impact of digital content consumption (e.g., lower framerates/resolutions) or decreasing the amount of content consumed should be the main focus in areas with a carbon intensive energy mix, while countries with a cleaner energy mix could instead focus on extending the lifetime of devices, reuse/refurbish and recycling (since the use phase is relative clean anyway). I am not sure how to incorporate this into your paper, however, as of now I feel that the discussion about the topic is insufficient right now.

Another comment I have which is related to this is for whom this "global average consumption" – calculated to "40% of the per capita carbon budget" – is relevant, when the electricity mix ranges from 0.02 kg CO₂-eq kWh⁻¹ to 1.34 kg CO₂-eq kWh⁻¹? I feel the regional differences are so great that the applicability of the global average diminishes. This shows in lines 177-187 where you state that, on average, the consumption can increase or decrease by a certain percentage. In reality, it really depends on where we are looking. In many places the difference is negligible. As I said, this can be a problematic conclusion but it is nonetheless important to emphasize that certain changes (both structural and behavioral) are mainly relevant for some context. Perhaps the solution is to problematize your own conclusions/findings more.

The number of internet users worldwide increases steadily (from approximately 40% in 2015 to 60% in 2020), and especially in India (20% in 2018, 40% in 2020). This will obviously impact the carbon budget, especially since the growth of internet users occur mainly in places with a carbon intense energy mix. You calculate the population growth until 2050 but I can't find anywhere that you consider the increased adoption rate. Does this impact your estimated carbon budget for digital content consumption in any way?

In lines 19-20, you say that "potentially a transition to less energy-intensive devices and settings is paramount to prevent the growing Internet demand from hindering sustainable lifestyles" – I am not sure I agree. Yes, in some contexts it might be better from a purely environmental point of view to switch to more energy efficient hardware, but in many countries with a relatively clean energy mix it is usually better to keep using the same hardware for as long as possible (especially if we take into consideration other sustainability-related factors related to the extraction, manufacturing and disposal of the equipment).

In your reply to a previous comment, you stated that "some impacts have a strong regional component, like those connected to nutrients, toxic, and particulate matter emissions linked to the mining and processing of metals for electronic devices", and indeed this is certainly the case. You added lines 308-315 as a response which I think works, however, what I was really asking for was for you to emphasize that digital content consumption in one place does not only lead to "global" effects (such as climate change) but also to environmental/social effects in completely different places (mainly global North behaviors leading to global South problems – waste, pollution, etc.) I think the section can be reworked to clarify that.

Lines 54-59 I feel you are underselling your contributions here. It's a bit too general, and it feels like you're basically stating the obvious ("making everything better and more sustainable is the best"). Try to make this bit more engaging.

Response to Reviewers for NCOMMS-23-04500A

We extend our gratitude to the Reviewers for their time and dedicated efforts in evaluating the revised version of our article. We appreciate their positive feedback regarding the enhancements introduced in this new iteration of our work. It is indeed heartening to note that Reviewer #1 views our manuscript as approaching readiness for publication, while Reviewer #4 acknowledges the ongoing improvements. Their positive assessments provide us with great encouragement to address the remaining concerns and issues and to deliver a version suitable for potential publication in Nature Communications. We remain deeply committed to enhancing our work, and we have taken the Reviewers' insightful comments into careful consideration, implementing the needed changes to ensure their suggestions are effectively incorporated into the manuscript.

The format adopted is as follows: comments in blue, replies in black, and actions in **bold**. Page and line numbers refer to the main article with highlighted changes.

Reviewer #1

Thanks for the opportunity to review the article again. The authors have adequately addressed all my comments from the first round.

Overall, the article is now much clearer. The article still makes an original contribution by putting the footprint of digital consumption in relation to the earth's carrying capacity.

We are pleased to see that our article has improved after the first revision and that the approach of contextualizing the environmental footprint of digital content consumption in relation to the Earth's carrying capacity is considered to be a valuable contribution.

I am aware that the analysis and future scenarios are subject to many uncertainties. The authors have addressed the most critical uncertainties and the overall results seem plausible.

We thank to the Reviewer's for acknowledging the improvements in our analysis of the future scenarios. In response to Reviewer #2's valuable input, we have refined this new version of the article by limiting the prospective analysis to the near-term, focusing on the time horizon up to 2030. This adjustment aligns with Reviewer #2's viewpoint that extending such analysis over several decades increases uncertainty. It is important to note that this change does not impact the primary findings of the prospective analysis.

However, there is one point I want to raise: When you provide country-specific results in line 88+, you consider differences in electricity mixes across countries. I assume regional differences in actual digital content consumption would also impact the results but have not been accounted for, as you focus on the "global average user". I am aware that considering diverging consumption levels across countries would add a lot of additional complexity. You mention this issue in the discussion section. However, by using a global average, many countries in the Global South are unfairly credited with high digital consumption that doesn't exist in reality. If these countries now have a CO₂-intensive electricity mix, they perform very poorly in a country comparison. Is there any way to make a rough estimate (even just an example) to address this uncertainty?

The Reviewer is right in noting that the country-specific results only consider differences in electricity mixes across countries. As mentioned in the article (page 3, line 61), the digital content consumption patterns, encompassing factors such as the type of digital content consumed, time dedicated to each digital content, or type of devices used, are representative for the global average user across all Internet users. While we acknowledge the large heterogeneity in Internet users across countries, it is noteworthy that data on consumption patterns is indeed scarce and not readily available in official reports or peer-reviewed papers. In our study, we sourced this information from online platforms specialized in market and consumer data, as comprehensively documented in Supplementary Tables 2 and 3.

Recognizing that our ability to make accurate assumptions about country-specific consumption patterns is constrained by these data limitations, we sought to partially address this issue through the sensitivity analysis presented in Supplementary Figs. 6 and 7. More specifically, Supplementary Fig. 6 explores the impacts for a user archetype who only uses a smartphone –the end-user device with the lowest electricity intensity–, watches video streaming at low quality (480p resolution), listens music at low quality, and joins online meetings with audio only. In this case, the most critical impacts could decrease by 53–67% relative to the average user. On the other hand, Supplementary Fig. 7 outlines the impacts of a user who consumes content at the highest quality, demonstrating the potential for increased impacts by 21–31%.

In response to the Reviewer’s concerns, **we have implemented several changes to enhance the significance of our findings, clarifying in the Results section that the country-specific assessment focuses on electricity mix differences across countries (page 4, line 89).** Additionally, **we have expanded the Discussion section to incorporate the results of the sensitivity analysis for the two hypothetical user archetypes. This not only acknowledges the potential impact of consumption levels variations but also underscores the importance of improving data availability on user consumption patterns for more comprehensive and geographically sensitive studies (page 15, line 310).**

Other than that, the article is ready for publication, in my opinion.

We sincerely appreciate the Reviewer's positive feedback during the review process.

Reviewer #2

SECOND REVIEW OF “THE ROLE OF DIGITAL CONTENT CONSUMPTION IN ENVIRONMENTALLY SUSTAINABLE LIFESTYLES” (REVIEWER #2)

There are two main issues raised by reviewers that the responses of the authors fail to capture. The first is that it is absurd to estimate energy and emissions for information technology (IT) to 2050. The second (raised by Reviewer 4 and with which I concur) is that this article frames the issues around emissions of IT as a problem of individual behavior rather than a problem that requires systemic response.

We acknowledge and sincerely regret the ongoing concerns expressed regarding our analysis as presented in the revised manuscript. However, we are also pleased to see that the Reviewer finds our work useful and appreciate the value of our results on the nearer term environmental impacts of digital content consumption. In this regard, we have taken these concerns into careful consideration and made significant efforts to address both issues, as outlined below.

On the first key issue, the authors can do all the sensitivities they want, and state that they are just exploring alternative futures, but it's just not possible to say anything sensible about ICT electricity use and emissions two or three decades from now. Imagine someone in the year 2000 trying to predict what would happen in the next 23 years. Could they have predicted the build out of data center infrastructure from 2000 to 2005, the rise of the Internet of things, the dominance of handheld computing (phones), the emergence of hyperscale data centers, or the recent rush to machine learning? I don't think so. Things change too fast in IT, and I don't think it helps when researchers who apparently don't understand this issue try to create scenarios for several decades hence. Ultimately this is an issue for the editor to decide, but if it were up to me, I wouldn't publish research that embodies the flawed idea that projecting several decades hence makes any sense in this context.

We acknowledge the challenges associated with projecting ICT electricity use and emissions several decades into the future, given the rapid evolution of the IT sector. We understand such an approach can raise concerns on the accuracy of the prospective environmental impacts of digital content consumption. In response to the Reviewer's observations, we have made substantial modifications to the section “Opportunities for mitigating the impacts of digital content consumption” in order to address these concerns:

1. **We have limited the prospective analysis to the near-term, up to 2030, as recommended by the Reviewer.** This modification aligns with the Reviewer's viewpoint that extending such analysis over several decades increases uncertainty. **The revisited results, presented in Figure 5, focus on the nearer term and allow us to emphasize two key findings: (i) that a rapid decarbonisation of the global power sector by 2030 could substantially reduce the GHG emissions associated with ICT (as these emissions come mainly from electricity consumption during the manufacturing and operation of electronic devices), and (ii) that concerns persist regarding the use of mineral and metal resources (page 9, line 194).** It is important to note that the only prospective element considered in the results presented in Figure 5 concerns the future electricity generation scenarios. **We have revisited the text as well as the caption of Figure 5 to ensure that we clearly communicate this assumption (page 9, line 184).**
2. Having identified that the decarbonisation of the power sector might not be sufficient to achieve a holistic reduction across all the studied categories, **we additionally present a sensitivity analysis to examine the potential benefits of extending the lifetime of electronic devices. The results, presented in Figure 6, demonstrate that extending devices lifetime can significantly reduce the use of mineral and metal resources (page 11, line 222).** For instance, doubling the lifetime of electronic devices has the potential to decrease the per capita carrying capacity from 55% to 29%, considering the current electricity generation scenario, or from 60% to 32% when considering the 1.5 °C-aligned 2030 scenario.
3. Finally, **we have removed the second part of the prospective analysis that involved projections of ICT-related parameters, such as energy efficiency (Fig. 6 in the revised manuscript).** This step was taken in recognition of the increased uncertainty associated with projecting such parameters compared to electricity mix projections.

On the second main issue, the title of the article tells the tale: “The role of digital content consumption in environmentally sustainable lifestyles”. It’s not about content consumption and it’s not about lifestyles, it’s about a system that people use to apply IT to their lives. To first order, people’s choices of which content to consume DO NOT MATTER to the electricity use and emissions associated with the IT system, and it is a mistake to imply that this is fundamentally a problem of consumption and lifestyles. For example, the energy use of routers, switches, and other network equipment doesn’t change much when these devices are either running flat out or not being used at all. The energy use of most computers is dominated by standby and off power, which are also not affected much by consumption or lifestyle choices. So the framing of this article is all wrong.

While we believe that individual preferences and choices may have an influence on the environmental outcome (e.g., through the type of devices used), we recognize that the energy use and emissions linked to the IT system are predominantly influenced by systemic factors. Following the Reviewer’s suggestion, we have implemented the following changes to avoid any potential misinterpretation:

1. **We propose to change the title to "On the environmental sustainability of digital content consumption".**
2. **We have carefully reviewed the Abstract, eliminating references to user consumption behaviour and lifestyles, thereby focusing on technological opportunities for impact mitigation, notably the decarbonisation of the power sector and the extension of electronic devices lifetimes (page 1, line 16).**
3. **Similarly, we have removed the analysis and discussion related to the influence of user’s preferences on the environmental outcome, both in the section “Opportunities for mitigating the impacts of digital content consumption” and in the Discussion section. Instead, our analysis now exclusively emphasises technological opportunities for impact mitigation (page 9, line 184 and page 13, line 257).**

These changes have allowed us to reframe our work in a manner consistent with the systemic nature of the issue at hand. We believe these revisions address the concerns raised by the Reviewer.

I think the authors have done some useful work and could say something important about nearer term questions. If they just focused on 2030 instead of 2050 and moved away from implying that ICT emissions are mainly a subject for individual consumption and lifestyle choices, then I could get behind publishing it. But as it stands, I recommend that it not be published in its current form.

We believe that implemented revisions address the Reviewer's concerns and provide a more robust and focused analysis of the near-term environmental impacts of digital content consumption as well as the systemic approach to impacts mitigation. We hope these changes address the core issues raised.

Reviewer #4

I would first like to thank the authors for the very detailed responses to our feedback on your initial draft. It makes our work much easier and shows that you have put effort and consideration into this new version of the manuscript. Overall, I feel that your manuscript is improving, although I still have some general comments that I would like you to consider in your next round of revisions.

We would like to express our gratitude to the Reviewer for your thoughtful review of our revised manuscript. We are glad to hear that the article is improving and have taken the new comments into careful consideration and have made appropriate changes to incorporate them.

While I value the notion of assigning individuals an annual carbon budget for allocation across diverse ICT-related activities, along with the breakdown of carbon consumption linked to specific ICT applications like video streaming, I do acknowledge the presence of certain inherent constraints within this approach. These acknowledged limitations underscore the need for careful consideration. For example, as I also emphasized last time, it implies that a Norwegian user uses much less of their carbon budget to consume digital content than an Indian user. You clearly describe this in the paper, but I feel that there are problematic implications of this fact are not properly discussed. Either it implies that Norwegian (or Western in general) users should be “allowed” to consume much more digital content than for example Indian users (by a great deal, too). This means that reducing the ICT usage in developing countries would be beneficial for the climate, but in many developing countries (such as Norway or Sweden) the climate impact of these activities is almost negligible. This is the harsh reality as of right now, with the current electricity mix in these two contexts being radically different, but stands in stark contrast with your statement: “While it is crucial for all users to embrace more sustainable consumption habits and for the energy and ICT sectors to reduce emissions worldwide, the endeavour to ensure sustainable development can be approached through various principles. In practice, developed countries are likely to play a leading role and accelerate their decarbonisation efforts to compensate for the lower decarbonisation pace in developing countries”. Alternatively, it implies that we should in fact use different budgets for different contexts, but I am not sure how such a budget would be designed. In practice, focusing on either lowering the impact of digital content consumption (e.g., lower framerates/resolutions) or decreasing the amount of content consumed should be the main focus in areas with a carbon intensive energy mix, while countries with a cleaner energy mix could instead focus on extending the lifetime of devices, reuse/refurbish and recycling (since the use phase is relative clean anyway). I am not sure how to incorporate this into your paper, however, as of now I feel that the discussion about the topic is insufficient right now.

We greatly appreciate the Reviewer for pinpointing the potential implications of allocating the carbon budget on an equal per capita basis. The presence of these inherent constraints indeed highlights the complexity of this approach and calls for careful consideration. In light of these valuable insights, we have extended the discussion within the paper to address the Reviewer's concerns adequately.

As we have acknowledged in the paper, the allocation principle to share the carbon budget is a much-debated topic, and several principles have been proposed in the literature, including equality, needs, right to development, sovereignty, and capability principles. In this work, we have used the equality or equal share per capita principle for several reasons. First, this principle is in alignment with the principles of Sustainable Development that recognise the equal rights to resources of past, current, and future Earth populations. Secondly, this approach has been recommended as the best approach as it works in isolation from complex justice-related and ethical considerations [1]. Finally, the equality principle is the most common approach in the literature, indicating a general acceptance among researchers (despite the lack of global consensus on the topic). Moreover, we note that *“any broadly accepted way of going beyond the “equal share per capita” approach is currently lacking”* [2]. **We have now revisited the Results section to incorporate this rationale for choosing the equality principle (page 5, line 17).**

Having said that, we understand the concerns raised by the Reviewer regarding the practical implications of using the equality principle. It is right that the results may suggest that users in regions with a carbon intensive electricity mix should consider reducing their consumption to align with sustainability targets. However, a more nuanced understanding, suggested by the Reviewer, reveals the importance of context-specific mitigation

strategies. While it remains imperative for both the energy and ICT sectors to reduce emissions worldwide, countries where the use phase has less environmental impact due to a cleaner electricity mix might focus on strategies aimed at diminishing embodied impacts, including extending the lifetime of electronic devices and recycling. Conversely, in regions with carbon-intensive electricity mixes, the greatest potential for mitigation lies in the urgent decarbonization of their electricity production. This outcome carries significant implications in the context of the anticipated widespread adoption of the Internet by a substantial portion of the approximately 3 billion unconnected people in the coming decades. Notably, the majority of these unconnected individuals reside in countries strongly reliant on fossil fuels reliant countries such as India, China, Pakistan, and Nigeria. **We have now revisited the Discussion section to include the context-specific mitigation discussion (page 13, line 274). Moreover, we now acknowledge that “utilising alternative principles that rely on differentiated responsibilities would result in higher carbon budgets allocated to users in developing countries. Accordingly, the footprint of digital content consumption would become less relevant in developing countries with larger ecological budgets, and more critical in developed countries with smaller allocated environmental shares.”, while highlighting the lack of consensus regarding a broadly accepted principle beyond the equal per capita share (page 14, line 290).**

[1] Ryberg, M. W., Andersen, M. M., Owsianiak, M. & Hauschild, M. Z. Downscaling the planetary boundaries in absolute environmental sustainability assessments – A review. *Journal of Cleaner Production* 276, 123287 (2020)

[2] Dao, H., Peduzzi, P. & Friot, D. National environmental limits and footprints based on the Planetary Boundaries framework: The case of Switzerland. *Global Environmental Change* 52, 49–57 (2018).

Another comment I have which is related to this is for whom this “global average consumption” – calculated to “40% of the per capita carbon budget” – is relevant, when the electricity mix ranges from 0.02 kg CO₂-eq kWh⁻¹ to 1.34 kg CO₂-eq kWh⁻¹? I feel the regional differences are so great that the applicability of the global average diminishes. This shows in lines 177-187 where you state that, on average, the consumption can increase or decrease by a certain percentage. In reality, it really depends on where we are looking. In many places the difference is negligible. As I said, this can be a problematic conclusion but it is nonetheless important to emphasize that certain changes (both structural and behavioral) are mainly relevant for some context. Perhaps the solution is to problematize your own conclusions/findings more.

Regional variations are indeed significant in the context of diverse electricity mixes. In this context, we believe that the global average is useful to provide a broad overview of the impacts of digital content consumption. Yet, we acknowledge the need to highlight the importance of regional disparities. **Based on the previous comment, we have addressed this concern by expanding the Discussion section to delve into the regional variability of impacts and its implications for mitigation efforts. This also includes further discussion around the potential implications of the equal per capita sharing principle (page 13, line 274).**

The number of internet users worldwide increases steadily (from approximately 40% in 2015 to 60% in 2020), and especially in India (20% in 2018, 40% in 2020). This will obviously impact the carbon budget, especially since the growth of internet users occur mainly in places with a carbon intense energy mix. You calculate the population growth until 2050 but I can't find anywhere that you consider the increased adoption rate. Does this impact your estimated carbon budget for digital content consumption in any way?

We appreciate the Reviewer for bringing up this important aspect. In our analysis, the increasing rate of Internet adoption does not directly affect the per capita carbon budget. This is because the global carbon budget is distributed among the entire global population, rather than being allocated solely to Internet users.

In lines 19-20, you say that “potentially a transition to less energy-intensive devices and settings is paramount to prevent the growing Internet demand from hindering sustainable lifestyles” – I am not sure I agree. Yes, in some contexts it might be better from a purely environmental point of view to switch to more energy efficient hardware, but in many countries with a relatively clean energy mix it is usually better to keep using the same hardware for as long as possible (especially if we take into consideration other sustainability-related factors related to the extraction, manufacturing and disposal of the equipment).

We appreciate this comment, which aligns with the perspective shared by Reviewer #2 on adopting a systemic approach to address the environmental impacts of ICT infrastructure. As suggested, **we have carefully revised the Abstract and the Results sections to move away from the mitigation potential of user consumption behavior and lifestyles. Our focus now centers on technological opportunities for impact mitigation, particularly the decarbonization of the power sector and the extension of electronic device lifespans (page 1, line 16).**

In your reply to a previous comment, you stated that “some impacts have a strong regional component, like those connected to nutrients, toxic, and particulate matter emissions linked to the mining and processing of metals for electronic devices”, and indeed this is certainly the case. You added lines 308-315 as a response which I think works, however, what I was really asking for was for you to emphasize that digital content consumption in one place does not only lead to “global” effects (such as climate change) but also to environmental/social effects in completely different places (mainly global North behaviors leading to global South problems – waste, pollution, etc.) I think the section can be reworked to clarify that.

We thank the Reviewer for clarifying their concerns regarding the regional environmental/social effects resulting from digital content consumption in different places. This is indeed at the core of the life cycle assessment approach, i.e., to identify potential burden shifting between regions. We regret that this aspect was not sufficiently clear in the first round of reviews.

In response to this concern, **we have reworked the Discussion section to explicitly acknowledge that specific environmental impacts linked to the extraction and processing of raw materials used in electronic devices have a strong regional component, such as those related to nutrients emissions, ecotoxicity impacts, and land use. We further highlight that these impacts tend to occur in geographic locations distinct from where digital content consumption takes place (page 13, line 262).**

Lines 54-59 I feel you are underselling your contributions here. It’s a bit too general, and it feels like you’re basically stating the obvious (“making everything better and more sustainable is the best”). Try to make this bit more engaging.

Thank you for pointing out this issue. **We have revised the conclusions paragraph in the Introduction section to provide more specific and engaging results from our study. In particular, we now highlight that “a rapid decarbonisation of electricity could reduce the climate impacts of digital content consumption to only 12% of the per capita carrying capacity as soon as by 2030”, while acknowledging that “concerns surrounding the use of mineral and metal resources may persist, even with extended electronic device lifetimes”. Finally, we emphasise that “it is essential that roadmaps towards a sustainable ICT sector adopt a more holistic perspective that goes beyond mitigating energy consumption impacts by encompassing measures focused on significantly reducing the extraction of new raw materials for electronic devices.” (page 2, line 51).** We believe that these revisions have made the text more engaging and responsive to your feedback.

Reviewers' Comments:

Reviewer #2:

Remarks to the Author:

The authors have addressed my concerns in the latest draft.

Reviewer #4:

Remarks to the Author:

Thanks again for thorough responses to my comments last time.

The introduction is now clear and it is focusing on summarizing the current state and clearly presents your findings.

One thing related to the results that I might have mentioned earlier. I am wondering if the "primary reasons for internet usage among users" (row 60) correlates with the most energy consuming/CO2 emitting activities? I was thinking about video gaming for example which in many cases (especially PC gaming) requires much more advanced hardware and consumes more power than for example social media. Since you're focusing on energy consumption and emissions and not time spent online, it would be more relevant to check those activities that contribute most to energy consumptions and emissions. I know these might be the same activities, but it's not clear.

(Row 164-167) Another thing that I am wondering about is how the embodied emissions are calculated. Clearly the average lifetime of a smartphone, for example, is much shorter than that of a desktop. The desktop I am using have outlived at least three or maybe four smartphones (obviously this is just an anecdote). I cannot find how these things are taken into consideration in your calculations, but if you have thought about this then please ignore this comment.

253-256 - Is it actually EQUALLY important? The transportation and food systems contribute much more to climate change and other negative impacts than ICT

310-324 Some of this is more related to method than discussion in my perspective, but it's not objectively wrong to have this discussion here either.

Other than those small comments I think that the manuscript should soon be considered publishable.

Title: On the environmental sustainability of digital content consumption

RESPONSE TO REVIEWERS

The format adopted is as follows: comments in **blue**, replies in black, and actions in **bold**. Page and line numbers refer to the main article with highlighted changes.

Reviewer #4

Thanks again for thorough responses to my comments last time.

The introduction is now clear and it is focusing on summarizing the current state and clearly presents your findings.

We express our gratitude to the Reviewer for evaluating the revised version of our article and for the positive comments provided. We are happy to see that the changes made in the introduction section, prompted by the previous round of comments, have improved clarity. We have carefully considered the remaining comments by the Reviewers and provide a point-by-point response below.

One thing related to the results that I might have mentioned earlier. I am wondering if the "primary reasons for internet usage among users" (row 60) correlates with the most energy consuming/CO2 emitting activities? I was thinking about video gaming for example which in many cases (especially PC gaming) requires much more advanced hardware and consumes more power than for example social media. Since you're focusing on energy consumption and emissions and not time spent online, it would be more relevant to check those activities that contribute most to energy consumptions and emissions. I know these might be the same activities, but it's not clear.

Response: Thank you for emphasizing this important aspect. We recognize that video gaming may demand more advanced hardware and consume more power. However, integrating gaming into our analysis poses challenges as it does not consistently involve Internet consumption, which is the primary focus of our study. During our research, we encountered a lack of readily available statistics on gaming patterns (e.g., online versus offline gaming time for the average user) to model this aspect accurately, leading us to omit it from the analysis. Similarly, due to the lack of data, we have not considered other activities and services which require Internet, such as IoT devices used for automation in smart homes. We acknowledge this limitation in Supplementary Section 4. Since we left such contributions out of the study, our impact values represent a lower bound on the impacts that would be obtained including gaming and other omitted activities. Consequently, if the lower bound impacts are already relevant, the same will hold for the *true* impacts. Moreover, our modelling choice avoids the large uncertainties that would be brought about by poor data on such activities.

To clarify this assumption and the scope of our analysis, we have added the following paragraph to the Methods section under Goal and scope definition:

- **Page 16, line 344: *"Additional activities and services dependent on the Internet, such as Internet of Things (IoT) devices or video gaming, have been omitted. However, their consideration would amplify the overall impacts making them more severe than what was considered herein. Hence, we report lower bounds on the "true" impacts that could be determined if all data on digital consumption were available."***

(Row 164-167) Another thing that I am wondering about is how the embodied emissions are calculated. Clearly the average lifetime of a smartphone, for example, is much shorter than that of a desktop. The desktop I am using have outlived at least three or maybe four smartphones (obviously this is just an anecdote). I cannot find how these things are taken into consideration in your calculations, but if you have thought about this then please ignore this comment.

Response: Thanks for this relevant comment. Indeed, we calculated the embodied emissions taking into account the specific lifetime of electronic devices. As outlined in the Methods section, *"Infrastructure inventory data for*

end-user devices and CPE were normalized per active hour based on the total amount of active hours over their operating lifetime (Supplementary Table 6). On the other hand, infrastructure inventory data for the IT equipment in data centres (i.e., servers and storage equipment) were normalized per GB of data traffic based on the global stock of servers and storage capacity in data centres, the outbound data traffic, and an average equipment lifetime (Supplementary Table 7)". Essentially, we have distributed the embodied emissions over the lifetime of the device, following a common approach in LCA. As noted by the Reviewer, different devices imply different lifetimes, and we have accounted for this, as shown in Supplementary Table 6. For example, in our analysis, a smartphone has an average lifetime of 3 years, while a TV has a lifetime of 10 years.

Recognizing that the calculation approach might not be fully clear in the manuscript, we have revisited the Methods to clarify that 'infrastructure inventory data' refers to embodied emissions as follows:

- **Page 17, line 380: "The impact embodied in end-user devices and CPE were normalized per active hour considering the total amount of active hours over their operating lifetime (Supplementary Table 5). On the other hand, the impact embodied in data centres IT equipment (i.e., servers and storage equipment) were normalized per GB of data traffic based on the global stock of servers and storage capacity in data centres, the outbound data traffic, and an average equipment lifetime (Supplementary Table 7)."**

253-256 - Is it actually EQUALLY important? The transportation and food systems contribute much more to climate change and other negative impacts than ICT

Response: We acknowledge the Reviewer observation that transportation and food systems contribute much more to climate change impacts than ICT. Our intended message is that while addressing transportation and food systems is crucial, it's also important not to overlook the potential impact of ICT in efforts to reduce consumption impacts.

We have revised the sentence accordingly:

- **Page 13, line 259: "The food system and transport are typically identified as priority areas to reduce these impacts within the safe zone^{41,65-68}, but our findings highlight the importance of not overlooking efforts to mitigate the impacts of the ICT infrastructure to avoid exacerbating the pressure on the finite Earth's carrying capacity."**

310-324 Some of this is more related to method than discussion in my perspective, but it's not objectively wrong to have this discussion here either.

Response: We appreciate the Reviewer's suggestion and have made adjustments to address their comment. Specifically, we relocated the explanation of the definition of the two hypothetical users to the corresponding Supplementary Section 1.4 and Supplementary Figs. 6 and 7. In the main manuscript, the text now focuses on conveying the primary findings as follows:

- **Page 15, line 318: "To illustrate this point, we have considered two hypothetical users representing two extreme consumption patterns, finding that the most critical impacts could decrease by 53-67% or increase by 21-31% relative to the average user (see Supplementary Section 1.4 and Supplementary Figs. 6 and 7)."**

Other than those small comments I think that the manuscript should soon be considered publishable.

Response: Thanks for this positive comment. We are happy that the manuscript is now deemed suitable for publication after we made the proper changes.